



# Retrieval of Desert Dust and Carbonaceous Aerosol Emissions over Africa from POLDER/PARASOL Products Generated by GRASP Algorithm

Cheng Chen[1], Oleg Dubovik[1], Daven K. Henze[2], Tatyana Lapyonak[1], Mian Chin[3], Fabrice Ducos[1], Pavel Litvinov[4], Xin Huang[4], Lei Li[1]

[1]Laboratoire d'Optique Atmosphérique (LOA), UMR8518 CNRS, Université de Lille1, Villeneuve D'ASCQ, 59655, France
[2]Department of Mechanical Engineering, University of Colorado, Boulder, Colorado, 80309, USA
[3]NASA Goddard Space Flight Center, Greenbelt, Maryland, 20771, USA
[4]GRASP-SAS, Remote Sensing Developments, Université de Lille1, Villeneuve D'ASCQ, 59655, France

*Correspondence to*: Cheng Chen (cheng1.chen@ed.univ-lille1.fr) and Oleg Dubovik (oleg.dubovik@univ-lille1.fr)

**Abstract.** Understanding the role atmospheric aerosols play in the earth-atmosphere system is limited by uncertainties in the knowledge of their distribution, composition and sources. In this paper, we use the GEOS-Chem based inverse modelling
framework for retrieving desert dust (DD), black carbon (BC) and organic carbon (OC) aerosol emissions simultaneously from aerosol data retrieved from the polarimetric POLDER/PARASOL Aerosol Optical Depth (AOD) and Aerosol Absorption Optical Depth (AAOD) produced with the GRASP algorithm (hereafter PARASOL/GRASP). First, the inversion framework is validated in a series of numerical tests conducted with synthetic PARASOL-like data. These show that the framework allows for retrieval of the distribution and strength of aerosol emissions. For example, the uncertainty of retrieved
daily emissions in error free conditions is bellow 25.8% for DD, 5.9% for BC and 26.9% for OC. In addition, BC refractive index is sensitive to BC emission retrieval, which could produce an additional about 1.8 times differences for total BC emission. The approach is then applied to one-year (December 2007 to November 2008) of data over the African and Arabian Peninsula region using PARASOL/GRASP spectral AOD and AAOD at six wavelengths (443, 490, 565, 670, 865 and 1020 nm). Analysis of the resulting retrieved emissions indicates 1.8 times overestimation of the prior DD online
mobilization and entrainment model. For total BC and OC, the retrieved emissions show a significant increase of 209.9%~271.8% in comparison to the prior GEOS-Chem inventory of carbonaceous aerosol emissions. The model posterior simulation with retrieved emissions shows good agreement both with the AOD and AAOD PARASOL/GRASP products used in the inversion. The fidelity of the results is evaluated by comparison of posterior simulations with measurements from AERONET that are completely independent of and more temporally frequent than PARASOL observations. To further test
the robustness of our posterior emissions constrained using PARASOL/GRASP, the posterior emissions are implemented in the GEOS-5/GOCART model and the consistency of simulated AOD (prior: R=0.77, RMSE=0.14, MAE=0.09; posterior: R=0.81, RMSE=0.10, MAE=0.06) and AAOD (prior: R=0.65, RMSE=0.019, MAE=0.014; posterior: R=0.69,





RMSE=0.015, MAE=0.011) with other independent measurements (MODIS and OMI) demonstrates promise in applying this database for modelling studies.

## 1 Introduction

Atmospheric aerosols have a variety of sources and complex chemical compositions. Desert dust (DD) aerosol is one of

the most abundant types of aerosol by mass, while the range of global dust emission estimates span a factor of about five (Huneeus et al., 2011). Primary carbonaceous aerosol, which consists of black carbon (BC) and organic carbon (OC) from combustion of fossil fuels, biofuels and biomass, has strong light absorption that can affect the energy balance of the earth-atmosphere system. High uncertainty in carbonaceous aerosol emissions (e.g., Bond et al., 2004) translates into a significant high uncertainty in evaluating their climate effects (Textor et al., 2006). The Intergovernmental Panel on Climate Change

(IPCC) estimates the global mean direct radiative forcing due to primary carbonaceous aerosol of -0.1 w/m$^2$ in their 2001 report, in 2007 they raise it to 0.18 w/m$^2$, in the latest report (IPCC, 2013) the value comes to 0.31 w/m$^2$ (Myhre et al., 2013). Furthermore, desert dust and carbonaceous aerosols can have deleterious impacts on regional air quality and public health (Chin et al., 2007; Monks et al., 2009; Li et al., 2013). Thus, observations are needed to accurately evaluate their emissions in order to better understand the role atmospheric aerosols play in the earth-atmosphere system (Bellouin et al., 2005).

Space-borne remote sensing instruments offer an integrated atmospheric column measurement of the amount of light scattering by aerosols through modification of diffuse and direct solar radiation. Numerous satellite observations of the spatial and temporal distribution of aerosols have been conducted in the last two decades (King et al., 1999; Kaufman et al., 2002; Lenoble et al., 2013). The satellite retrievals of Aerosol Optical Depth (AOD) and Aerosol Absorption Optical Depth (AAOD) are directly related to light extinction and absorption due to the presence of aerosol particles. AOD is a basic

optical property derived from many earth-observation satellite sensors, such as AVHRR (Advanced Very High Resolution Radiometer), MODIS (Moderate Resolution Imaging Spectroradiometer), MISR (Multi-angular Imaging SpectroRadiometer) and POLDER (Polarization and Directionality of the Earth's Reflectances) (Goloub et al., 1999; Geogdzhayev et al., 2002; Kahn et al., 2009; Tanré et al., 2011; Levy et al., 2013). AAOD is another valuable product to quantify the solar absorption potential of aerosol; however, only a few satellite sensors can provide retrievals of AAOD, and

only with limited accuracy, for example OMI (Ozone Monitoring Instrument) on the Aura satellite (Torres et al., 2007; Veihelmann et al., 2007) and POLDER on PARASOL (Polarization & Anisotropy of Reflectances for Atmospheric Sciences coupled with Observations from a Lidar), because only ultraviolet (UV) and shortwave visible channels and polarimetric measurements are sensitive to aerosol absorption.

Despite their ability to provide a high-degree of spatial coverage, satellite measurements alone are not sufficient for

answering question regarding the distributions, magnitudes, and fates of aerosols in the atmosphere. These aspects can be studied using Chemical Transport Models (CTMs), which incorporate meteorological data from external databases with atmospheric physics, considering the physical and chemical processes in the atmosphere, and allow modelling of detailed





distribution of aerosol for any chosen time period (e.g. models by Balkanski et al., 1993; Chin et al., 2000; Takemura et al., 2000; Ginoux et al., 2001; Bessagnet et al., 2004; Grell et al., 2005; Spracklen et al., 2005; Mann et al., 2010) . However, CTM simulations are limited by uncertainties in knowledge of aerosol emissions characteristics, knowledge of atmospheric processes, and the meteorological data used. As a result, even the most recent models are found to capture only the principal

global features of aerosol. For example, among different models, quantitative estimates of average regional aerosol properties often disagree by amounts exceeding the uncertainty of remote sensing of aerosol observations (Chin et al., 2002, 2014, Kinne et al., 2003, 2006; Textor et al., 2006). Therefore, there are diverse and continuing efforts to harmonize and improve aerosol modelling by refining the meteorology, atmospheric process representations, emissions and other components (e.g. aerosol aging scheme, particle mixing state etc.) (Watson et al., 2002; Dabberdt et al., 2004; Generoso et

al., 2007; Ghan and Schwartz, 2007; He et al., 2016; Wang et al., 2014a, 2016).

One of the most promising approaches for reducing model uncertainty is to improve the aerosol emission fields (that is input for the models) by inverse modelling, i.e. fitting satellite observations and model estimates and by adjusting aerosol emissions. For example, Dubovik et al. (2008) developed an algorithm for inverting MODIS data and implemented the approach to retrieve distributions of aerosol emissions. The algorithm was used to implement the first formal retrieval of

global emission distributions of fine mode aerosol from the MODIS fine mode AOD data. Wang et al. (2012) and Xu et al. (2013) use MODIS radiances to constrain aerosol sources over China. Huneeus et al. (2012, 2013) optimize global aerosol emissions from MODIS AOD with a simplified aerosol model (Huneeus et al., 2009). However, as discussed in works such as Dubovik et al. (2008) and Meland et al. (2013), MODIS AOD (as well as currently available aerosol satellite data) contains only limited information to evaluate aerosol types, properties, or speciated emissions. Further, inconsistencies

between representations of aerosol microphysics between the CTM and the aerosol retrieval algorithm can have significant influences on inverse modelling of aerosol sources (e.g. Drury et al., 2010; Wang et al., 2010). Therefore, the retrieval of aerosol emissions from satellite observations remains very challenging.

The recently generated PARASOL/GRASP (General Retrieval of Atmosphere and Surface Properties) spectral AOD and AAOD data (Dubovik et al., 2011, 2014; data available from ICARE data distribution portal: http://www.icare.univ-

lille1.fr/) present new opportunities for constraining DD, BC and OC sources because their optical properties vary dramatically in the spectrum of short wave visible to near infrared (VIS-NIR) viewed by PARASOL. Polarimetric remote sensing measurements such as those from PARASOL have been postulated to provide much greater constraints on speciated aerosol emissions and microphysical properties (Meland et al., 2013). DD aerosols are dominated by coarse mode particles, and their AOD varies slightly in the VIS-NIR spectral range; in contrast, the AOD of fine mode dominated BC and OC

aerosols decrease sharply in this spectral range. In addition, DD and OC particles absorb most strongly in the UV and short wave visible channels, such as 443 nm, while BC particles are absorbing more ubiquitously (Sato et al., 2003). The GRASP retrieval overcomes the difficulty of deriving aerosol over bright surfaces in the shortwave visible wavelengths, which should help improve constraints of DD emissions over source regions, rather than having to rely on downwind observations (e.g., Wang et al., 2012).



Here we develop an inverse modelling approach to retrieve the spatial and temporal distributions of DD, BC and OC aerosol emissions simultaneously from PARASOL/GRASP spectral AOD and AAOD using the GEOS-Chem model (Bey et al., 2001) and its adjoint (Henze et al., 2007). Section 2 describes the model and data used in this study. The dust and carbonaceous aerosol model in the GEOS-Chem adjoint of Henze et al. (2007) is that of the GOCART (Goddard Chemistry

Aerosol Radiation and Transport) model implemented in GEOS-Chem (Fairlie et al., 2007; Park et al., 2003), which is fully conceptually consistent with the aerosol model used in the inversion by Dubovik et al. (2008). The details of inverse modelling and performance evaluation of the inversion framework using numerical tests are presented in the Section 3.  In order to interpret the retrieval results and improve our understanding of aerosol emissions, we retrieve one-year of daily DD, BC and OC emissions (see the Section 4). Evaluation of these inversion results using independent AERONET, MODIS and

OMI observations, as well as implementation of the posterior emissions in the GEOS-5/GOCART model is presented in Section 5. Conclusions and discussion of the study's merits and limitations are considered in the Section 6.

## 2 Model and data description

### 2.1 Study Area

The study area (30°W-60°E, 40°S-40°N) is shown in Figure 1, which covers the whole of Africa and the Arabian

Peninsula, comprising one of the largest dust source and biomass burning regions of the globe. The spatial and temporal variability of DD, BC and OC aerosols in this area has drawn numerous research (Duncan et al., 2003; Prospero and Lamb, 2003; Engelstaedter et al., 2006; Liousse et al., 2010; Zhao et al., 2010; Ginoux et al., 2012; Ealo et al., 2016). The number of PARASOL/GRASP retrievals per 0.1° x 0.1° grid box over a year (December 2007 to November 2008) and 28 AERONET (AErosol RObotic NETwork) (Holben et al., 1998) sites used to evaluate GEOS-Chem model simulations and

PARASOL/GRASP retrievals are shown in Figure 1. Note that the GRASP algorithm performs aerosol retrievals at PARASOL's native resolution of 6~7 km; each 0.1° grid box could thus have more than one GRASP retrieval, so the number of PARASOL/GRASP retrievals exceeds the number of days in some grid boxes of Figure 1. The amount of GRASP algorithm (see in section 2.3) retrievals over Northern Africa Sahara and the Arabian Peninsula desert region is relatively high, whereas other regions have a reduced number of retrievals due to the presence of clouds.



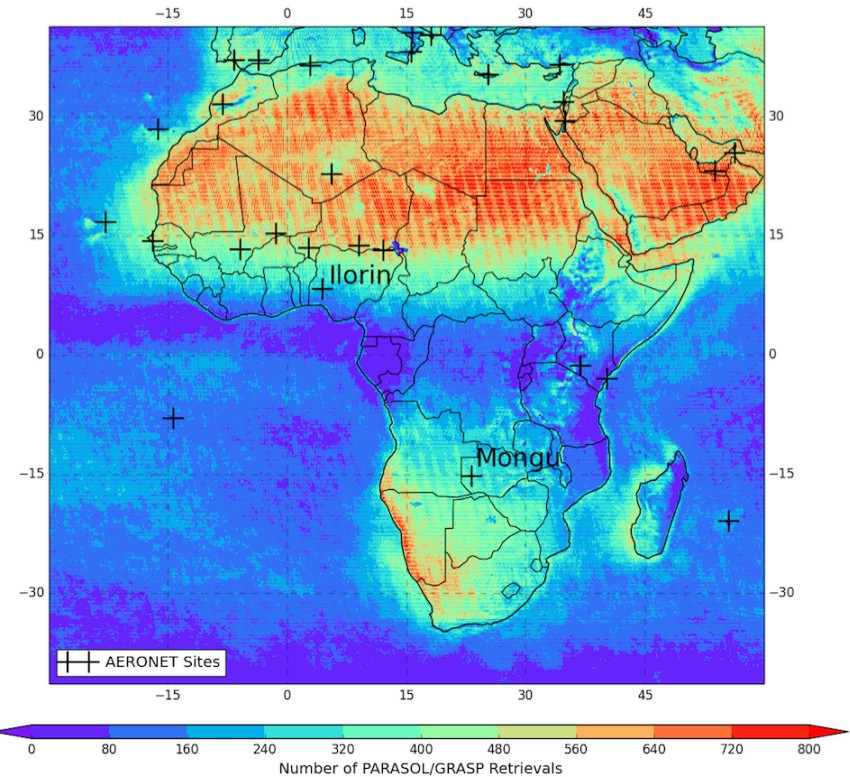

**Figure 1.** Distribution of PARASOL/GRASP AOD retrievals per 0.1° x 0.1° grid cell over a year (December 2007 to November 2008); the 28 AERONET sites used for validation are also shown with black cross.

### 2.2 GEOS-Chem model and its adjoint

5        GEOS-Chem is a global 3-dimensional chemical transport model driven by assimilated meteorological data from the NASA Goddard Earth Observing System Data Assimilation System (GEOS-DAS) (Bey et al., 2001). We use the GEOS-Chem (v9-02) model for forward aerosol simulation with 47 layers vertical resolution and 2° (latitude) x 2.5° (longitude) horizontal resolution. DD, BC and OC aerosols are simulated in our study, including 7 size bins for resolving dust (Fairle et al., 2007), and total aerosol mass of black and organic carbon (Park et al., 2003). Dust simulations in GEOS-Chem (Fairlie et

10  al., 2007) consist of the mineral dust entrainment and deposition (DEAD) model (Zender et al., 2003), which is coupled with the GOCART dust source function (Ginoux et al., 2001). The daily biomass burning sources are calculating from version 3 of Global Fire Emissions Database (GFED) inventory (van der Werf et al., 2006, 2010; Randerson et al., 2013). The monthly



anthropogenic fossil fuel and biofuel BC and OC emissions are adopted from Bond inventory with base year 2000 (Bond et al., 2007). The sulphate (SU) and sea salt (SS) aerosol simulation in GEOS-Chem is described in (Park et al., 2004; Jaeglé et al., 2011). The standard aerosol dry deposition in GEOS-Chem is described in (Wang et al., 1998; Wesely, 1989), and accounting for gravitational settling and turbulent mixing of particles to the surface (Pye et al., 2009; Zhang et al., 2001).

Aerosol wet deposition is through wet scavenging in convective updrafts as well as in- and below-cloud scavenging from convective and large scale precipitation (Liu et al., 2001).

The GEOS-Chem model assumes external mixing for all aerosol components with lognormal size distributions. The modal diameter and width for each dry aerosol species and their optical properties is specified. The extinction and scattering coefficients are calculated from size distributions and refractive indices assuming spherical particles. Different aerosol

species are considered to have hydroscopic growth rates as a function of ambient relative humidity (RH). The simulated aerosol masses are then converted to AOD ($\tau$) and AAOD ($\tau_a$) through the general relationship between aerosol optical depth and aerosol mass (Tegen and Lacis, 1996):

$$\tau(\lambda) = \sum_{i=1}^{n} \frac{3 Q_{ext,i}(\lambda)}{4 \rho_i r_{e,i}} m_i \tag{1}$$

$$\tau_a(\lambda) = \sum_{i=1}^{n} \frac{3 Q_{abs,i}(\lambda)}{4 \rho_i r_{e,i}} m_i \tag{2}$$

where $n$ is the total number of aerosol components, $i$ represents individual aerosol component, $m$ is the aerosol mass, $\lambda$ is wavelength, $\rho$ is aerosol particle density, $r_e$ is the particle effective radius, and $Q_{ext}(\lambda)$ and $Q_{abs}(\lambda)$ are the aerosol particle

extinction and absorption coefficients, respectively. The size distribution and the spectral aerosol refractive index used to calculate $Q_{ext}(\lambda)$ and $Q_{abs}(\lambda)$ are assumed based on Global Aerosol Data Set (Koepke et al., 1997), with modifications for dust particles by including a spectral dependence for the imaginary part based on analysis of AERONET measurements (Dubovik et al., 2002b). Further, the particle optical properties $Q_{ext}(\lambda)$ and $Q_{abs}(\lambda)$ are calculated according to AERONET Kernel based on mixture of spheroids suggested in studies by Dubovik et al. (2002a, 2006). The particle density and

hydroscopic growth rate are described in Chin et al. (2002) and Martin et al. (2003). Table 1 lists the detailed aerosol properties used in this study. Here we consider two cases of BC refractive index. Figure 2a demonstrates the relative humidity dependence of these two cases of BC aerosols extinction ($Q_{ext}(\lambda)/r_e$) at 565 nm, and Figure 2b presents the wavelength dependence of single scattering albedo (SSA) for these two cases. The Case 1 BC refractive index is based on Chin et al. (2002) and Martin et al. (2003). More recent studies have recommended a BC refractive index of 1.95-0.79i

(Schuster et al., 2005; Bond and Bergstrom, 2006; Koch et al., 2009; Arola et al., 2011), which has higher absorption and scattering ability than Case 1. Figure 2a shows that the extinctions calculated from AERONET Kernel for Case 2 BC particle are about a factor 1.5 higher than for Case 1. The difference of SSA is small (Case 2 is about 2% higher at 565 nm when RH=0%), however the difference increases when RH=95%, for which Case 2 is about 18% lower at 565 nm. Since the particle absorption efficiency $Q_{abs} = (1 - SSA) \cdot Q_{ext}$, the Case 2 BC particle shows a higher absorbing ability than Case 1.



Sensitivity tests are conducted to evaluate how these two BC refractive indices influence the total BC emissions retrieval in Section 3.2.4. The aerosol spectral extinction and absorption coefficients used in this study are all available online (http://csuchencheng.wixsite.com/chencheng/research-blog). It should be noted that the particle morphologies can affect the computation of scattering and absorption properties (Liu and Mishchenko, 2007; Mishchenko et al., 2013). However, usually

CTMs use external mixture of different aerosol components as described above for GEOS-Chem model used in present studies. The inclusion of more adequate internal mixing rules for calculating optical properties of resulted aerosol is a crucial for further improvements in CTMs aerosol simulation that is a subject for future developments. Indeed, since CTMs are aimed to account for all important chemical and physical transformations of aerosol particles. Therefore, in principle CTMs should provide all information about particles sizes and morphologies necessary for making adequate modelling of aerosol

optical properties. However, at the current stage the level of details in CTMs is not sufficient to model fully adequate component mixing and, as a result, the conversion from aerosol mass to aerosol optical properties is based on simplified external mixing rule size distributions and refractive indices known from in situ and remote sensing observations.

**Table 1.** Aerosol refractive index, size distribution and particle density for DD, BC, OC and host water employed in this
study

| Aerosol | Complex refractive index | | | Size distribution (μm) | | | Density |
|---|---|---|---|---|---|---|---|
| | n | k(440/670/870/1020) | | $r_{mean}$ | $r_{eff}$ | sigm | (g/cm$^3$) |
| DD | 1.56 | 0.0029/0.0013/0.0001/0.0001 | DST1 | 0.0421 | 0.14 | 2.0 | 2.5 |
| | | | DST2 | 0.0722 | 0.24 | 2.0 | 2.5 |
| | | | DST3 | 0.1354 | 0.45 | 2.0 | 2.5 |
| | | | DST4 | 0.2407 | 0.80 | 2.0 | 2.5 |
| | | | DST5 | 0.4212 | 1.40 | 2.0 | 2.65 |
| | | | DST6 | 0.7220 | 2.40 | 2.0 | 2.65 |
| | | | DST7 | 1.3540 | 4.50 | 2.0 | 2.65 |
| BC (Case 1) | 1.75 | 0.45 | | 0.0118 | 0.039 | 2.0 | 1.0 |
| BC (Case 2) | 1.95 | 0.79 | | 0.0118 | 0.039 | 2.0 | 1.0 |
| OC | 1.53 | 0.005 | | 0.0212 | 0.087 | 2.2 | 1.8 |
| Water | 1.33 | 1.0e-8 | | | | | 1.0 |





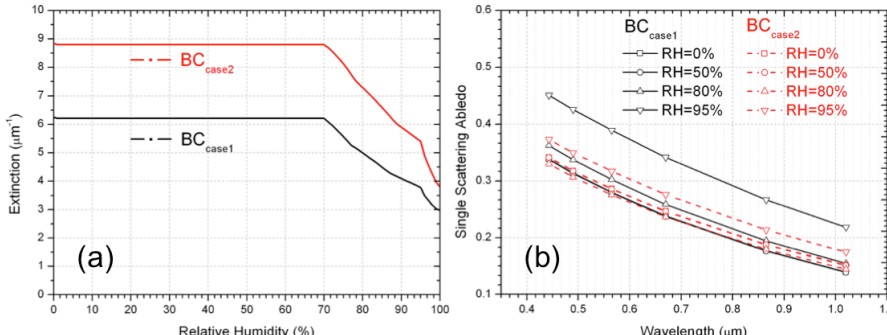

**Figure 2.** (a) The relative humidity dependence of BC particle extinction at 565 nm; (b) Wavelength dependence of BC particle single scattering albedo at six PARASOL wavelengths;

An adjoint model is an efficient tool for calculating the gradient of a scalar model response function with respect to a large set of model parameters simultaneously (Fisher and Lary, 1995; Elbern et al., 1997, 2000, 2007; Henze et al., 2004; Sandu et al., 2005). The adjoint of the GEOS-Chem model was developed specifically for inverse modelling of aerosols or their precursors and gas emissions (Henze et al., 2007, 2009). The 4D-Variational data assimilation technique is used to optimize aerosol emissions by combining observations and model simulations. The adjoint of GEOS-Chem has been widely

used to constrain emissions. For example, Kopacz et al. (2009) utilized MOPITT measurements of carbon monoxide (CO) columns to optimize Asian CO sources. Zhu et al. (2013) constrain ammonia emissions over the U.S. using TES (Tropospheric Emission Spectrometer) measurements. Zhang et al. (2015) use OMI AAOD to constrain anthropogenic BC emissions over East Asia. However, these studies have focused on a single aerosol or gas species and kept others constant during the inversion, since the satellites or other available observations of aerosols generally did not provide enough accurate

information to estimate contributions from different species. The recent development of the PARASOL/GRASP retrieval, which retrieves more detailed and accurate aerosol information (see in section 2.3), thus presents a new opportunity for constraining emissions from different aerosol species simultaneously, which has only been considered in few studies (e.g., Xu et al., 2013).

### 2.3 PARASOL/GRASP aerosol products

GRASP is a highly versatile and accurate aerosol retrieval algorithm that processes properties of aerosol and land surface reflectance. The algorithm is developed for enhanced characterization of aerosol properties from spectral, multi-angular polarimetric remote sensing observations (http://www.grasp-open.com/) (Dubovik et al., 2011, 2014; Lopatin et al., 2013). The POLDER/PARASOL imager provides spectral information of angular distribution of both total and polarized components of solar radiation reflected to space. With the expectation of 3 gaseous absorption channels (763, 765 and 710



nm), the observations over each pixel include total radiance at 6 channels (443, 490, 565, 670, 865 and 1020 nm) and linear polarization among 3 channels (490, 670 and 865 nm). The number of viewing angle is similar for all spectral channels and varies from 14 to 16 depending on solar zenith and geographical location. Meanwhile, PARASOL provides global coverage about every 2 days. Comprehensive measurements (~144 independent measurements per pixel) from PARASOL allow
GRASP to infer aerosol properties including spectral AOD and AAOD, the particle size distribution, single scattering albedo, spectral refractive index and the degree of sphericity (some description of GRASP aerosol products can be found in papers of Kokhanovsky et al. (2015) and Popp et al. (2016)). Extensive information of aerosol distribution and their properties provides a means to constrain specific aerosol types, which is vital to characterizing emissions from different aerosol species.

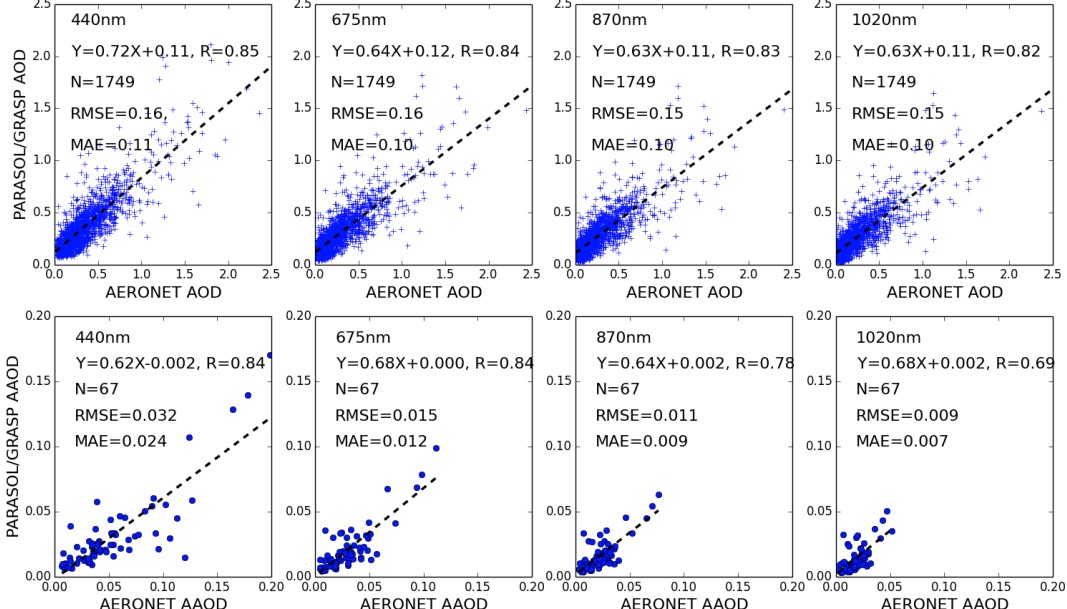

**Figure 3.** Validation of one-year of PARASOL/GRASP spectral AOD and AAOD rescaled to 2.0° x 2.5° horizontal resolution with AERONET 28 sites measurements at 440, 675, 870 and 1020 nm wavelengths over the study area; the number of matched pairs (N), correlation coefficient (R), root mean square error (RMSE) and mean absolute error (MAE) are provided in the top left corner.

In this study, we adopt one-year (December 2007 to November 2008) PARASOL products of spectral AOD and AAOD from GRASP to retrieve DD, BC and OC emissions over the study area in Section 4. In order to evaluate the reliability of PARASOL aerosol products from GRASP, we compared PARASOL/GRASP retrievals with AERONET



measured AOD and AAOD at 4 Sun photometer channels (440, 670, 870 and 1020 nm) in Figure 3. Here, we use level 2 AERONET data, which are cloud screened and quality assured (Smirnov et al., 2000). From all one-year measurements collected from 28 sites, we extract data between 13:00 p.m. and 14:00 p.m. local time. This provides a 60 minute window centered at the PARASOL overpassing time of ~13:30 p.m. The averaged AERONET sun-direct AOD and AAOD by

inversion of almucantar measurements (Dubovik et al., 2000; Dubovik and King, 2000) over this 60 minute window are averaged for comparison with PARASOL/GRASP retrievals. We aggregate the PARASOL/GRASP products into 2 ° latitude x 2.5° longitude horizontal resolution to match the spatial resolution used by GEOS-Chem; any 2° x 2.5° grid box with less than 500 available PARASOL/GRASP retrievals for averaging is omitted. Depending on geographical location, the number of GRASP retrievals in a single 2° x 2.5° grid box ranges from 500 to 1600. Figure 3 presents the validation of retrieved

PARASOL AOD and AAOD by GRASP algorithm against the AOD and AAOD measured by AERONET. There is a solid correlation between PARASOL/GRASP and AERONET for AOD as well as AAOD. For example, the correlation coefficients (R) are 0.85 and 0.84, and the root mean square errors (RMSE) are 0.16 and 0.032, and the mean absolute errors (MAE = $\frac{1}{N}\sum_{i=1}^{N}|(M_i - O_i)|$) are 0.11 and 0.024 for AOD and AAOD at 440 nm respectively.

## 3 Methodology

### 3.1 Description of inverse modelling

Our inverse modelling approach optimizes BC, OC and DD emissions at the 2° x 2.5° horizontal resolution of the forward GEOS-Chem model, driven by GEOS-5 meteorological fields with 6h temporal resolution. The algorithm iteratively seeks adjustments to emissions in order to minimize the differences between observations and simulations as quantified by

the cost function, $J$, given by the sum of following quadratic form:

$$J_{obs}(\mathbf{S}) + J_{a\,priori}(\mathbf{S}) = \frac{1}{2}\sum(\mathbf{f}(\mathbf{S}) - \mathbf{f}_{obs})^{\mathrm{T}}\mathbf{C}_{obs}^{-1}(\mathbf{f}(\mathbf{S}) - \mathbf{f}_{obs}) + \frac{1}{2}\gamma_r(\mathbf{S} - \mathbf{S}_a)^{\mathrm{T}}\mathbf{C}_a^{-1}(\mathbf{S}^p - \mathbf{S}_a) \qquad (3)$$

The first term characterizes the fitting of observation, where the vector $\mathbf{f}_{obs}$ is the vector of observed values used for inversion and $\mathbf{f}(\mathbf{S})$ is the vector of simulated values based on emission sources $\mathbf{S}$, while the vector $\mathbf{S}$ describes generally the four-dimensional distribution of emissions. $\mathbf{C}_{obs}^{-1}$ is the error covariance matrix of $\mathbf{f}_{obs}$. The second term is introduced to constrain retrieval and it indicates the agreement with *a priori* estimates $\mathbf{S}_a$ of the emissions. $\mathbf{C}_a^{-1}$ is the error covariance

estimate of *a priori* emissions. $\gamma_r$ is a regularization parameter. Indeed, in general the information content of observation is insufficient for unique retrieval of all parameters describing emissions, i.e. the problem is ill-posed and some *a priori* information is needed. In most applications "prior model" emission from bottom-up inventories $\mathbf{S}_a$ (i.e. standard model emissions) are used as *a priori* estimates of fundamentally unknown emissions.

The minimization of the quadratic form given by Eq. (3) can be obtained by steepest decent iterations:


$$S^{p+1} = S^p + \Delta S^p,$$

$$\Delta S^p = \nabla J_{obs}(S^p) + \nabla J_{a\,priori}(S^p) = \mathbf{K}_{obs}^T \mathbf{C}_{obs}^{-1} \Delta f^p + \gamma \mathbf{C}_a^{-1}(S^p - S_a), \tag{4}$$

where $\mathbf{K}_{obs}^T$ denotes matrix of Jacobians of observation characteristics $f$. Equations (3) and (4) are written using vectors and matrices, describing four-dimensional geophysical fields that are generally are very large. However, in practice neither transport models nor inverse modelling algorithms (if emissions retrieved at high resolution) explicitly utilize matrix and

vectors. The transport models are generally organized as routines calculating continuous (i.e. with relatively small time step) time series of the geo characteristic resulted from time integration. For example, calculations of corrections $\Delta S^p$ are obtained by running the adjoint model that directly produces the product of $\mathbf{K}_{obs}^T \mathbf{C}_{obs}^{-1} \Delta f^p$ without explicit calculation of the Jacobians. For example, for inversion of observations of aerosol mass, i.e. $f = M$, the computations of gradient $\nabla J^p(t, \mathbf{x})$ of cost function $J^p(t, \mathbf{x})$ using the adjoint model can be expressed as time integration operation (see derivations by Dubovik et

al. (2008)) as following:

$$\nabla J^p(t, \mathbf{x}) = \int_t^{t_0} T^\#(t', \mathbf{x})(\nabla J^p(t', \mathbf{x}) + C_{obs}^{-1} \Delta m^p(t', \mathbf{x}))(-\mathrm{d}t') + \gamma_r C_a^{-1}(s^p - s_a) \tag{5}$$

where

$$\Delta m^p(t, \mathbf{x}) = m_{obs}(t, \mathbf{x}) - \int_{t_0}^t T(t', \mathbf{x})(m(t', \mathbf{x}) + s^p(t', \mathbf{x})) \, \mathrm{d}t' \tag{6}$$

and $T$ represents transport operator. $T$ and $m$ are explicit functions of time $t$ and spatial coordinates $\mathbf{x} = (x, y, z)$. $T^\#(t, \mathbf{x})$ is the adjoint of transport operator of $T(t, \mathbf{x})$, (the adjoint operation is a transformation of continuous function equivalent to matrix transposition operations) that is composed of adjoints $T_i^\#(t, \mathbf{x})$ of the component processes $T_i(t, \mathbf{x})$:

$$T^\#(t, \mathbf{x}) = T_1^\# T_2^\# T_3^\# \dots T_{i-1}^\# T_i^\# \tag{7}$$

The above equations describe an approach to invert transport model based on the measurements of aerosol mass $M_{obs}$, which is the direct simulation parameter in the chemical transport model. In our analysis, the aerosol data fields is available only in the form of AOD and AAOD from the satellite measurements:

$$f = \tau(t, \mathbf{x}) = F(m(t, \mathbf{x}), \lambda, Q_{ext}, Q_{abs}, \dots) \tag{8}$$

where $F(\dots)$ is a function converting aerosol mass $m(t, \mathbf{x})$ to AOD and AAOD based on spectral characteristics $\lambda$, aerosol extinction $Q_{ext}$ and absorption $Q_{abs}$ coefficients, etc, see in Eq (1-2). The correction $\Delta S^p(\mathbf{x}) = \nabla J^p(t, \mathbf{x})$ minimizing the

form of Eq. (3) that relates for fitting of AOD and AAOD under *a priori* constraints can be written as:

$$\nabla J^p(t, \mathbf{x}) = \int_t^{t_0} T^\#(t', \mathbf{x}) F^\#(t', \mathbf{x})(\nabla J^p(t', \mathbf{x}) + C_{obs}^{-1} \Delta \tau^p(t', \mathbf{x}))(-\mathrm{d}t') + \gamma_r C_a^{-1}(s^p - s_a) \tag{9}$$

here $F^\#(t', \mathbf{x})$ is adjoint operator corresponding to matrix operation $\mathbf{F}^T$, where matrix $\mathbf{F}$ contains first derivatives $d\tau/dm$. It should be noted that GEOS-Chem adjoint model is developed for inversion of mass (or AOD at single wavelength), therefore the operator $F^\#(t', \mathbf{x})$ for inversion spectral AOD and AAOD was developed as a part of this work.



In principle, the methodology assumes that the *a priori* information is available, i.e. before the inversion, which here are the default model emissions. Unfortunately, the covariance matrix $\mathbf{C}_a$ of *a priori* emissions is not known accurately. As a result, this matrix is often assumed diagonal, where the elements of diagonal are equal or defined using rather simple strategies. Therefore, in order to address this fundamental lack of knowledge of $\mathbf{C}_a$, the contribution of the *a priori* term

(second term) in Eq. (3) is weighted by a regulation parameter, $\gamma_r$. This strategy is adapted here.

In addition, the GEOS-Chem adjoint model previously has been used for calculation of the gradient of Eq. (9) with respect to a vector of emissions scaling factors $\boldsymbol{\sigma}$ ($\boldsymbol{S}^p = \boldsymbol{S}_0 \boldsymbol{\sigma}^{p-1}$) (Henze et al., 2007). While the scaling factor formulation had the advantage of replace addition/subtraction correction of emissions (that can generate negative unphysical values) by division/multiplication of initial positive and non-zero $\boldsymbol{S}$, this can be realized in the inversion algorithm by transforming into

log scale (see discussion by Dubovik and King, (2000), Dubovik, (2004) and Henze et al. (2009)). However, the latter approach is rather challenging and GEOS-Chem uses empirically elaborated procedure (using equivalence ($\Delta \boldsymbol{S}/\boldsymbol{S} \sim \Delta ln(\boldsymbol{S})$)). Specifically, from the gradients of cost function with respect to aerosol emission scaling factors $\nabla_{\boldsymbol{\sigma}} J(t, \boldsymbol{x})$, the adjoint GEOS-Chem uses the L-BFGS-B optimization method (Byrd et al., 1995; Zhu et al., 1997), which affords bounded minimization of cost function, and ensuring positive values, to calculate the scaling factors for aerosol emissions. Figure 4 is the flowchart to

illustrate the methodology.

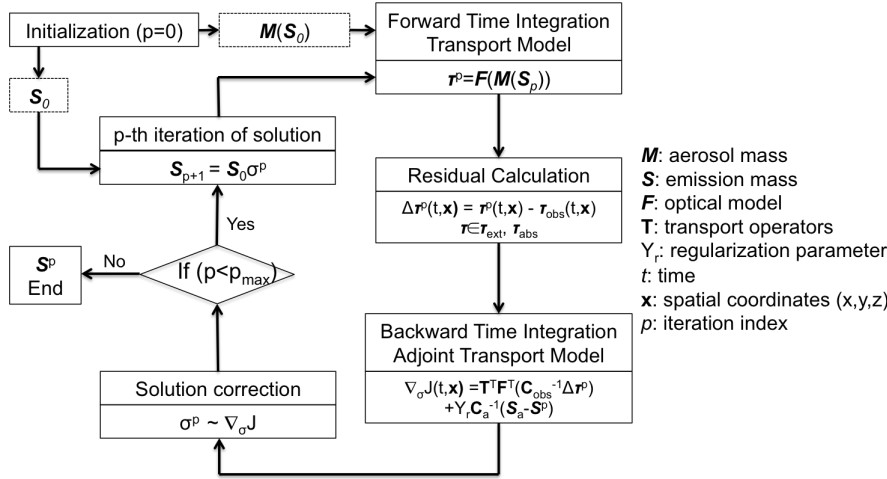

**Figure 4.** Diagram illustrating retrieval of aerosol emissions from satellite measurements

In order to optimize the specification of *a priori* constrains and initial guess, a number of synthetic tests were done in Section 3.2. It should be noted that using *a priori* estimate of emission $\boldsymbol{S}_0$ is not the only way of adding *a priori* constraints



in the inverse modelling. For example, Dubovik et al. (2008) demonstrated used of *a priori* knowledge on spatial and temporal variability of emissions, i.e. *a priori* limitation on derivative of corresponding functions (smoothness constraints). The potential advantage of smoothness constraints is that these limitations are milder than direct assumptions about values of emissions and therefore they introduce less systematic errors in the retrieval. However, such constraints are not used in this

study.

**3.2 Inversion test using synthetic measurements**

In this section, a series of numerical tests were performed to verify and illustrate how the algorithm inverts the synthetic measurements, and to tune the algorithm settings (e.g. initial guess, emission correction time resolution and BC refractive index). The retrieved results were compared with "True emissions". Synthetic measurements are PARASOL-like spectral

AOD and AAOD at six PARASOL wavelengths, simulated from 16 days of BC, OC and DD emissions, which, for simplicity, are specified to be constant over the 16 days, yet different from the prior model emissions in order to test the algorithm performance under the circumstances that *a priori* knowledge of the emission distribution is limited. Figure 5 shows the design of the inversion test from synthetic measurements.

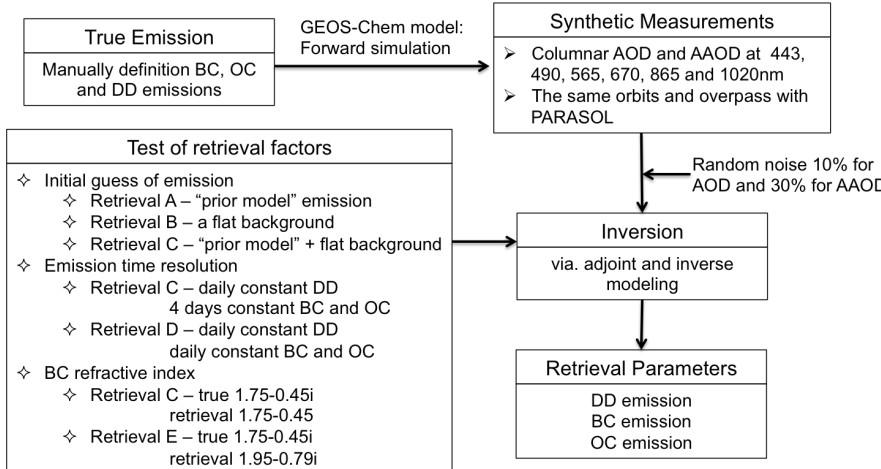

**Figure 5.** Diagram illustrating the inversion tests from synthetic measurements

**3.2.1 Spectrum weights**

In our inversion framework, the observed aerosol parameters contain AOD and AAOD at six PARASOL wavelengths. In principle, the weighting of observations of AOD and AAOD at these different wavelengths should be defined by the observation error covariance matrix $\mathbf{C}_{obs}$. For example, usually AOD is about ten times higher than the AAOD at the same

wavelength (SSA=1.0–AAOD/AOD) and therefore AAOD is expected to be retrieved and fitted more accurate on an





absolute scale. However, at present knowledge of this matrix is uncertain, so we thus perform the following sets of tests to optimize the observational covariance weights. The spectral residual values are defined to characterize the quality of spectral AOD and AAOD fit:

$$R_{AOD}(\lambda) = \sqrt{\frac{1}{N_i} \sum_{i=1,...,N_i} [\tau_{i,obs}(\lambda) - \tau_{i,model}(\lambda)]^2} \qquad (10)$$

$$R_{AAOD}(\lambda) = \sqrt{\frac{1}{N_i} \sum_{i=1,...,N_i} [\tau_{a,i,obs}(\lambda) - \tau_{a,i,model}(\lambda)]^2} \qquad (11)$$

The values of the spectral residuals $R_{AOD}(\lambda)$ and $R_{AAOD}(\lambda)$ are calculated after each iteration. The following options

were tested using well-known qualitative tendencies. In sensitivity test, two scenarios of spectrum weights are analysed. Since we are fitting absolute value of AOD and AAOD, the relative accuracy of retrieved AOD and AAOD ($\Delta\tau/\tau$ and $\Delta\tau_a/\tau_a$) are expected to be the same. The spectrum weights are defined as follows:

Option A, Unity weights for AOD and AAOD at 6 wavelengths: $[1,1,1,1,1,1]^T$ for AOD and $[1,1,1,1,1,1]^T$ for AAOD.

Option B, Unity weights for AOD but more weights on AAOD: $[1,1,1,1,1,1]^T$ for AOD and $[5,10,15,20,25,30]^T$ for AAOD.

The retrievals are conducted with option A and option B respectively, with other settings held constant. Comparison of spectral residuals after 20 iterations are shown in Figure 6, which indicates that Option B have a better fit for AAOD than Option A by increasing the weights for AAOD, although spectral AOD can be fitted comparably well using either option.

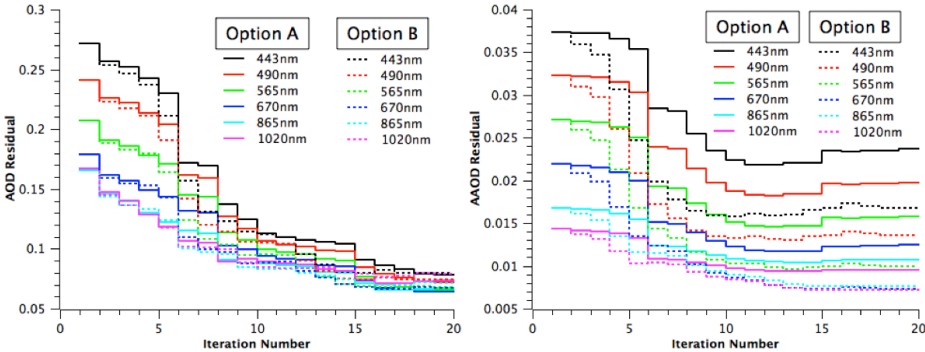

**Figure 6.** Comparison of spectral AOD and AAOD residual iteratively with two spectrum weight options

**3.2.2 Effect of initial guess in emission retrieval**

As mentioned in section 3.1, the emission retrieval is an ill-posed problem and utilization of *a priori* constraints and initial guesses are essential factors for the retrieval. In our retrieval framework, the emissions are adjusted using scaling factors that scaling for initial guess of emissions, $\boldsymbol{S}=\boldsymbol{S_0}\boldsymbol{\sigma}$. In principle, if the inverse problem is well-constrained the solution should be independent of the initial guess. Therefore, we analyse the dependence on initial guess using different retrieval





settings. The inversion is conducted with three different initial guess schemes that we describe in detail in the following sections. In each of these three schemes, the input synthetic measurements are 6 wavelengths AOD and AAOD, and the spectrum weights use the Option B scenario, while the retrieved emission correction time variations are assumed to be daily constant for DD and 4-day constant for BC and OC (note that we will separately test the assumption of emission correction

5    time resolution in section 3.2.3). Figure 7 shows the "True emissions" of DD, BC and OC and also the difference between true and retrieved emissions from three different initial guess schemes (Retrieval A, Retrieval B and Retrieval C). Figure 8 shows the scatter plots between BC, OC and DD emissions retrieved from Retrieval A, B, C versus true values.

A. Prior model: Initial guess is equal to Prior model emissions

10        In this method, the prior model emissions are directly used as the initial guess, therefore the adjustments of emissions are limited to the grid boxes with prior model emissions $S_0 > 0$. At the same time, the "True emissions" have difference with prior model. The upper panel in Figure 7 shows the assumed true BC, OC and DD emission distributions (units: kg/day) respectively. The second panel "Retrieval A - True" shows the differences between retrieved and true emissions from Retrieval A.



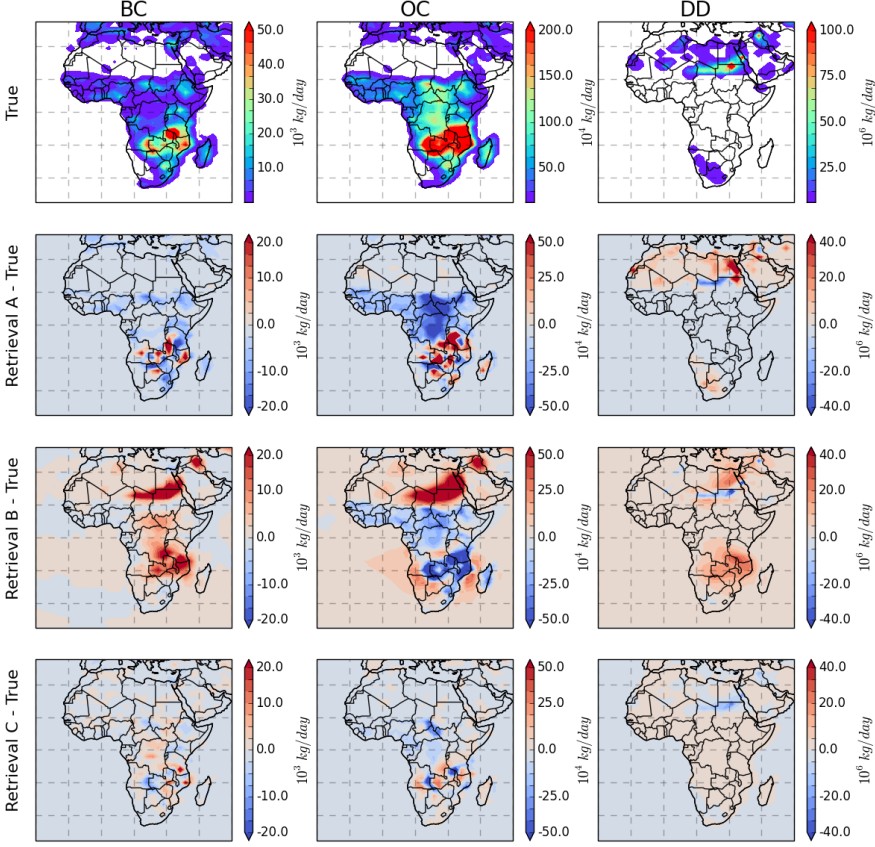

**Figure 7.** Inversion test for retrieving BC, OC and DD emissions from synthetic measurements with three different initial guess schemes: (A) Prior model emissions – Retrieval A; (B) Spatially uniform – Retrieval B; (C) Prior emission with spatially uniform background – Retrieval C;

For Retrieval A, the retrieval highly relies on the accurate distribution of model prior emissions, because the retrieval can only adjust the emissions on the grid boxes where the model prior emissions are non-zero, and thus the retrieval couldn't create new sources. In our inversion test, the model prior emissions are different from the truth both for distribution and strength. Therefore, as shown in Figure 8, the Retrieval A produces overestimations over the grid boxes that $S_0 > 0$, while





$S_{\text{true}}$=0, here $S_{\text{true}}$ represent true emissions, however the underestimations occur over the grid boxes that $S_0$=0, while $S_{\text{true}}$>0.

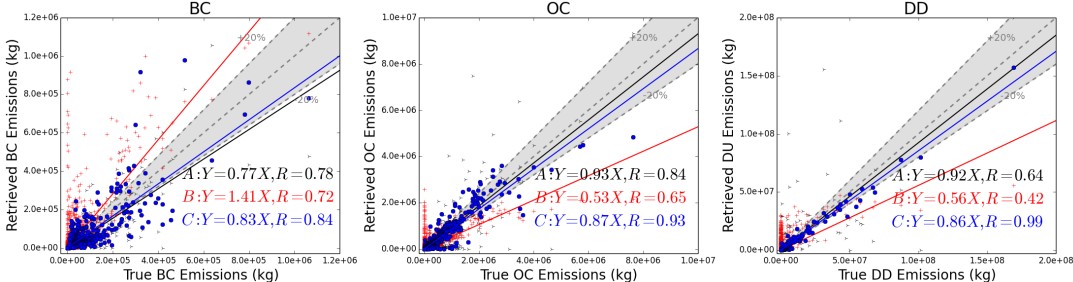

**Figure 8.** Scatter plots between BC, OC and DD emissions retrieved from Retrieval A, B, C versus true values

**B. Flat background everywhere**

For Retrieval B, we investigate the use of spatially uniform initial guesses for the emissions. With this initialization, we allow BC, OC and DD emission to be generated everywhere over land and ocean, which is equivalent to not using *a priori* knowledge of aerosol emissions. From the Figure 7 panel "Retrieval B – True", the algorithm can determine the intensive aerosol emission grid boxes, where high aerosol loading is observed. However, the desert dust and carbonaceous aerosol sources were not correctly reproduced since a uniform emission is used everywhere. The scatter plots between retrieved emissions from Retrieval B and true values are also shown in Figure 8. In this case, the retrieval could produce overestimations over some grid boxes where $S_{\text{true}}$=0. Although the uniform emission assumption gives the algorithm more freedom to find new sources, our tests indicate the retrieval could produce false sources in this assumption when the algorithm tries to determine BC, OC and DD emissions simultaneously. This misrepresentation indicates that the spectral AOD and AAOD are not sufficient to identify BC, OC and DD emission without any *a priori* knowledge.

**C. Prior model emission with flat background**

In retrieval C, the retrieval was initiated using prior model emissions but including a spatially uniform value over land grid boxes where $S_0$=0. In this study, the flat value equals to $10^{-4}$ Tg/day/grid for DD, $10^{-6}$ Tg/day/grid for BC, and $5\times10^{-6}$ Tg/day/grid for OC are used, which account for ~5% of the true emissions over entire area. This assumption allows retrieval of BC, OC and DD aerosol emissions everywhere over land (ship emissions over ocean are included in the model prior emissions), and at the same time it uses prior emission constraints to prevent false source generation. Figures 7 and 8 show that overall Retrieval C captures the emission distributions more accurately than Retrieval A and Retrieval B. The average ratio of retrieved emission to truth ($\sum_{N_{pixels}} \frac{S_{\text{retrieval}}}{S_{\text{true}}} / N_{pixels}$) for Retrieval C is $1.02 \pm 1.05$ for BC, $0.87 \pm 1.42$ for OC and $1.24 \pm 1.80$ for DD.



### 3.2.3 Assumption of emission correction time resolution

Aerosol sources are known to have high temporal and spatial variability. However, because PARASOL observations have limited temporal coverage (e.g. ~2 days global coverage, with observations once per day), the variability of aerosol emission at any given location can only be retrieved at a frequency no more than once per day. In order to investigate how assumptions regarding temporal variability of emission can affect the retrieval, we repeat the retrieval using two scenarios for emission correction: ET1, daily correction constant of DD, BC and OC emissions, and ET2, daily correction constant of DD emission and 4 days correction constant of BC and OC emissions. For each scenario, the input observations are 6 wavelengths of AOD and AAOD, and the retrieval is initialized by prior model emission with a uniform background emission (Retrieval C). Two scenarios are used. We test these two scenarios by conducting a 16-days retrieval, and Figure 9 shows the comparison between retrieved daily total DD, BC and OC emissions with the "True emissions". Note that the ET1 scenario uses the same settings with Retrieval C in section 3.2.2, and ET2 is named Retrieval D.


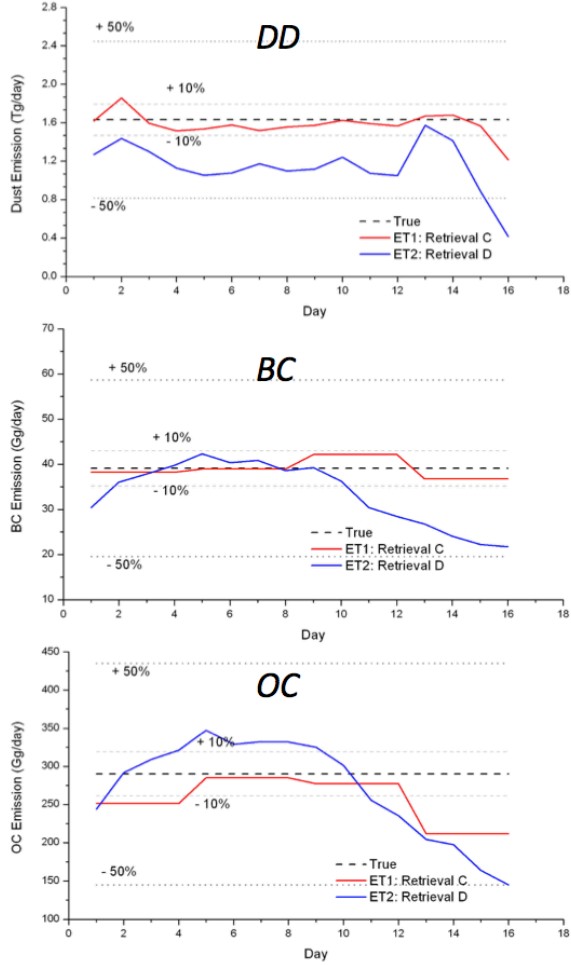

**Figure 9.** Sensitivity test for retrieving DD, BC and OC emissions over 16-days with two scenarios of assumption of emission correction time resolution

Figure 9 shows the retrieval maximum uncertainty ($\sum |S_{retrieval} - S_{true}| / \sum S_{true}$) for total daily DD emission over the study area is within 25.8% for Retrieval C, however this value reaches more than 50% for Retrieval D. For BC, the maximum uncertainty is within 5.9% for total daily emission from Retrieval C, while up to 40.8% for Retrieval D. The uncertainty of daily OC emission is within 26.9% for OC using Retrieval C, while about 38.6% for Retrieval D. Overall,





from this sensitivity test, the Retrieval C shows a better capability to capture the spatial distribution of DD, BC and OC emissions than Retrieval D, and it does not introduce false temporal variability.

### 3.2.4 Uncertainty in assumption of BC refractive index

Aerosol particles' light scattering and absorption efficiencies are determined by their complex refractive indices,
5   expressed as $m$= n-k$i$, where n is the real part and k is the imaginary part. The real part of the complex refractive indices defines the light scattering property of an aerosol species, whereas the imaginary part of the complex refractive indices determines the absorbing ability. Black carbon aerosol is the strongest atmospheric absorber of solar radiation. Its imaginary refractive index is at least about two orders of magnitude higher than other aerosol species (see Table 1). To identify the impact of the uncertainties of BC refractive index in our results, we test another commonly used specification of 1.95-0.79i
10   (Bond and Bergstrom, 2006) in our retrieval scheme (denoted as Retrieval E). Figure 10 compares the BC emission results from Retrieval E and Retrieval C (where the BC refractive index of 1.75-0.45i (Hess et al., 1998) was used) with the "true" BC emission.

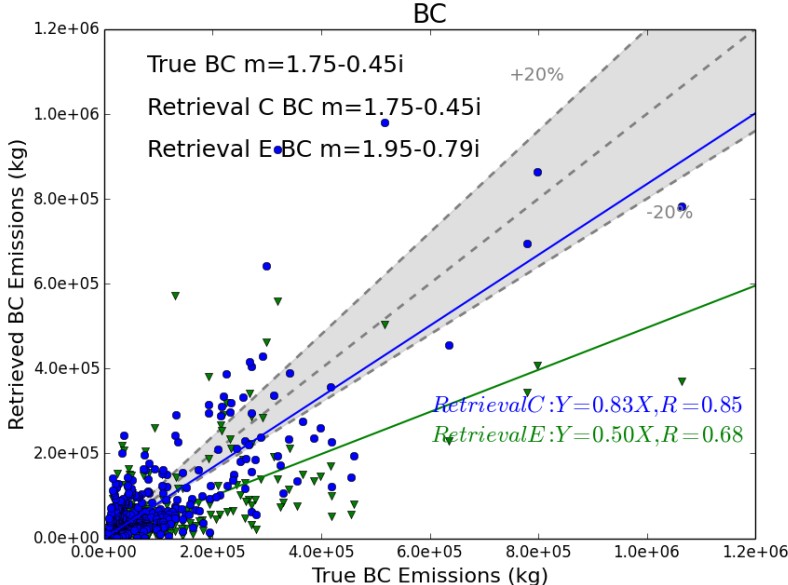

**Figure 10.** Test of BC particle refractive index influence on the retrieval of BC emissions.



The synthetic measurements of AOD and AAOD are simulated with BC refractive index $m$=1.75-0.45$i$, and the scenario Retrieval C uses the retrieval with the same BC refractive index; the slope of linear regression between the resulting retrieved and true BC emissions is 0.83, and the retrieved BC emission over study area is 39.0 Gg/day. In contrast, the Retrieval E scenario uses the retrieval with a higher BC absorption and scattering definition, $m$=1.95-0.79$i$, and as expected we get lower magnitudes of BC emissions (21.8 Gg/day), and the slope between retrieved and true BC emissions decreases to about 0.5. This sensitivity test demonstrates that uncertainty in the BC refractive index can lead to a factor of about 1.8 in total BC emissions.

Overall, these sensitivity tests show that our inversion scheme is capable of determining the strength and spatial distribution of BC, OC and DD emissions simultaneously from the multispectral PARASPL/GRASP AOD and AAOD products in the following manner:

1. Six wavelengths (VIS-NIR) AOD and AAOD from PARASOL/GRASP are needed to retrieve BC, OC and DD emissions simultaneously.

2. The improved spectral weighting factors for the PARASOL 6 wavelengths are $[1,1,1,1,1,1]^T$ for AOD and $[5,10,15,20,25,30]^T$ for AAOD, since it can provide a better fit of spectral AAOD.

3. The BC, OC and DD emissions are allowed everywhere over land. The retrieval is initialized by "prior model" emissions with a uniform background. The retrieval with this initialization could detect new sources and perform satisfactorily even when *a priori* knowledge of aerosol emission is not fully consistent with the assumed emissions.

4. The emission corrections are assumed daily constant for DD and 4 days constant for BC and OC. Owing to the limited observations available for assimilation, this assumption helps to make the retrieval more stable and accurate.

5. BC refractive index is sensitive to BC emission retrieval, which could produce a factor of ~1.8 differences between the two sets of commonly used BC refractive index data for total BC emission. We will produce two BC emission datasets with two scenarios of BC refractive index, Case 1: $m$=1.75-0.45$i$ and Case 2: $m$=1.95-0.79$i$.

## 4 Results

In this section, we discuss retrieval of DD, BC and OC emissions simultaneously from the actual PARASOL/GRASP spectral AOD and AAOD data from December 2007 to November 2008. The SU and SS aerosol simulations are kept as the prior model. PARASOL/GRASP retrievals were aggregated to the same horizontal resolution as the GEOS-Chem model (2° x 2.5°) and averaged within the grid cells prior to assimilation. When iteratively minimizing Eq. (3), the maximum iteration number was chosen to be 40, which takes about 60 days to complete on a computer workstation with 32x3.3 GHz CPUs.

### 4.1 Fitting of Aerosol Optical Depth

One of the important indicators of our inversion performance is the fitting of PARASOL/GRASP spectral AOD and AAOD. We evaluate the GEOS-Chem simulated spectral AOD at 443, 490, 565, 670, 865 and 1020 nm using prior or



posterior emissions against the corresponding PARASOL/GRASP retrieved AOD in Figure 11. The posterior GEOS-Chem spectral AOD are simulated using retrieved DD, BC and OC emissions, which will be presented in section 4.2. Figure 11a presents the annual average of the PARASOL spectral AOD from GRASP algorithm, whereas Figure 11b and 11c show the same quantity from the GEOS-Chem simulations with prior and posterior emissions, respectively. Here we extract GEOS-

Chem hourly AOD with the same PARASOL orbit partition at 13:00 p.m. local time, which is approximately the PARASOL overpass time of 13:30 p.m. Figure 11d and 11e display the grid-to-grid comparison between PARASOL/GRASP spectral AOD and prior and posterior GEOS-Chem simulation during one year, color-coded with the PARASOL Ångström exponent $\alpha_{443-865} = \frac{\ln(\tau_{443}/\tau_{865})}{\ln(865/443)}$. The Ångström exponent $\alpha$ is often used as a qualitative indicator of aerosol particle size; the smaller the $\alpha$, the larger the particle size. For example, the $\alpha$ values for "pure" dust aerosols are usually

near zero, whereas that for smoke or pollution aerosols are generally greater than 1 (Eck et al., 1999; Schuster et al., 2006).

One of the major discrepancies between the prior GEOS-Chem simulation and PARASOL/GRASP observation is that the model produces the highest annual average AOD values over the major dust source region of Northern Africa; however, satellite data show the maxima AOD in Central and the Southern Africa, where carbonaceous aerosols usually dominate (although Central Africa may also be influenced by dust events). Hence, compared to PARASOL/GRASP observations, the

prior GEOS-Chem AOD is overestimated in Northern Africa, while it is underestimated in the Southern Africa biomass burning and Arabian Peninsula regions. Some recent studies by Ridley et al. (2012, 2016) and Zhang et al. (2015) also indicate that the GEOS-Chem model overestimates dust AOD in Northern Africa. Meanwhile, Ridley et al. (2012) and Zhang et al. (2013) propose a new and realistic dust particle size distribution according to the measurements from Highwood et al. (2003), which can partially adjust the misrepresentation of dust near the source and over transport areas. This new

particle size distribution has been adopted in our prior and posterior GEOS-Chem simulation. In addition, the underestimation of model simulated AOD in biomass burning regions with the GFED emission database was also shown in other modeling studies (Chin et al., 2009; Johnson et al., 2016). The model simulated spectral AOD with the posterior emissions agree with the PARASOL observations much better, in spite of slight systematic overestimations from 565 nm to 1020 nm (about 13% on an annual average). This overestimation indicates some disagreement in modeling of AOD for these

bands that needs to be investigated and addressed in future studies.



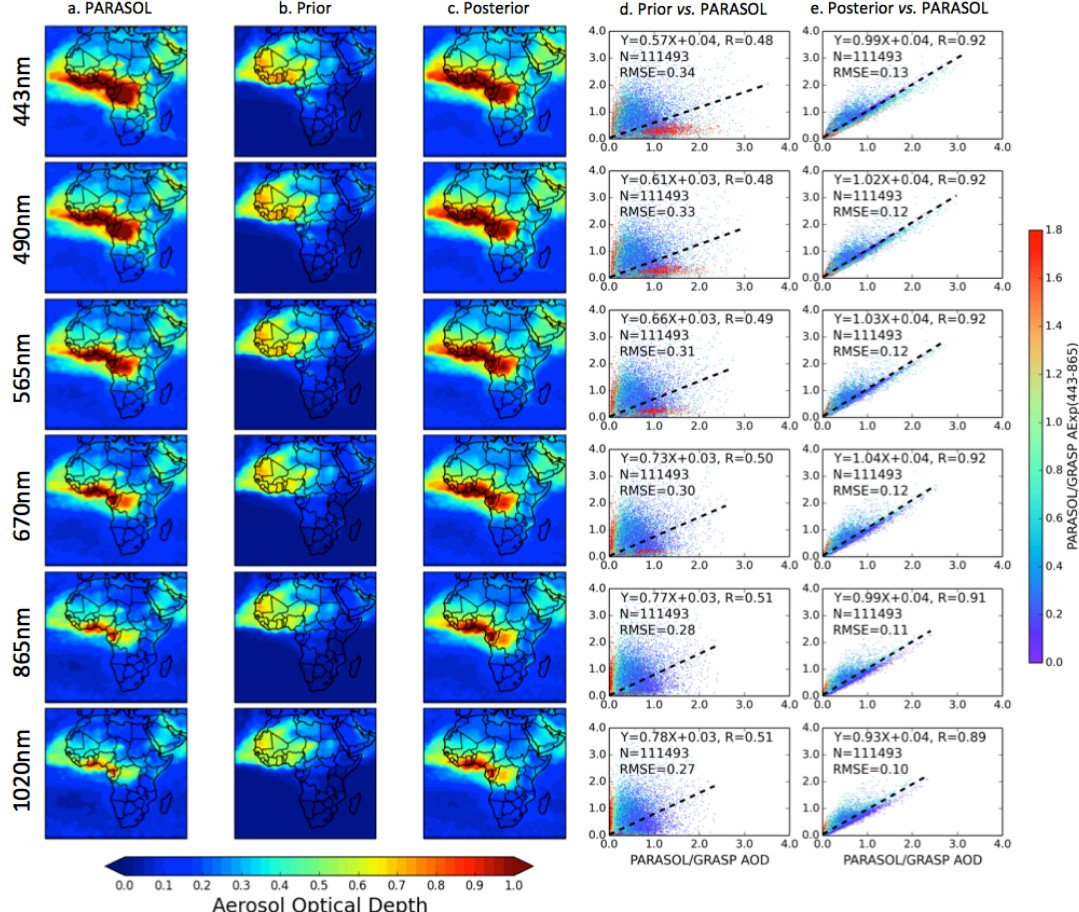

**Figure 11.** Comparison of the annual spatial distribution of prior (b) and posterior (c) GEOS-Chem simulated AOD at 443, 490, 565, 670 865 and 1020 nm with PARASOL/GRASP observations (a). The posterior spectral AOD are simulated using retrieved DD, BC and OC emissions. The scatter plots are grid-to-grid comparisons between PARASOL/GRASP spectral observations versus prior (d) and posterior (e) GEOS-Chem simulation during one year. The correlation coefficient (R) and root mean square error (RMSE) are provide in the top left corner.

Figure 11d and 11e show the statistics of prior and posterior GEOS-Chem simulated AOD versus PARASOL/GRASP observed AOD at 6 wavelengths during the entire year. The number of matched pairs is 111,493. For the GEOS-Chem



simulation with the posterior emissions, all the statistics parameters between model and observation are improved at all 6 wavelengths compared to the simulation with prior emissions. For example, the correlation coefficient has increased from 0.49-0.51 to 0.89-0.92 and the root mean square error has decreased from 0.27-0.34 to 0.10-0.13. Such improvements are expected as the posterior emissions are retrieved based on the PARASOL/GRASP AOD data. We will show further

5  evaluations with other datasets in Section 5.

**4.2 Fitting of Aerosol Absorption Optical Depth**

Similar to the AOD analysis, here we evaluate the fitting of AAOD (Figure 12). From the annual-averaged spectral AAOD in Figure 12, the prior GEOS-Chem simulation (Figure 12b) shows significant underestimations of AAOD over the entire domain compared to PARASOL/GRASP observations (Figure 12a). On the other hand, the posterior GEOS-Chem

10  simulation (Figure 12c) produces much better agreement with the PARASOL/GRASP data for all wavelengths, with a small overestimation of AAOD in the spectral range from 443 nm to 565 nm (about 6% on annual average) and a small underestimation at 865 nm and 1020 nm (about 9% on annual average). Linked with the ~13% overestimation of annual AOD from 565 nm to 1020 nm, this systematic phenomenon of fitting is possibly due to the model's relatively coarse resolution results in misrepresentations of DD, BC and OC emissions in some grid boxes.

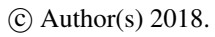



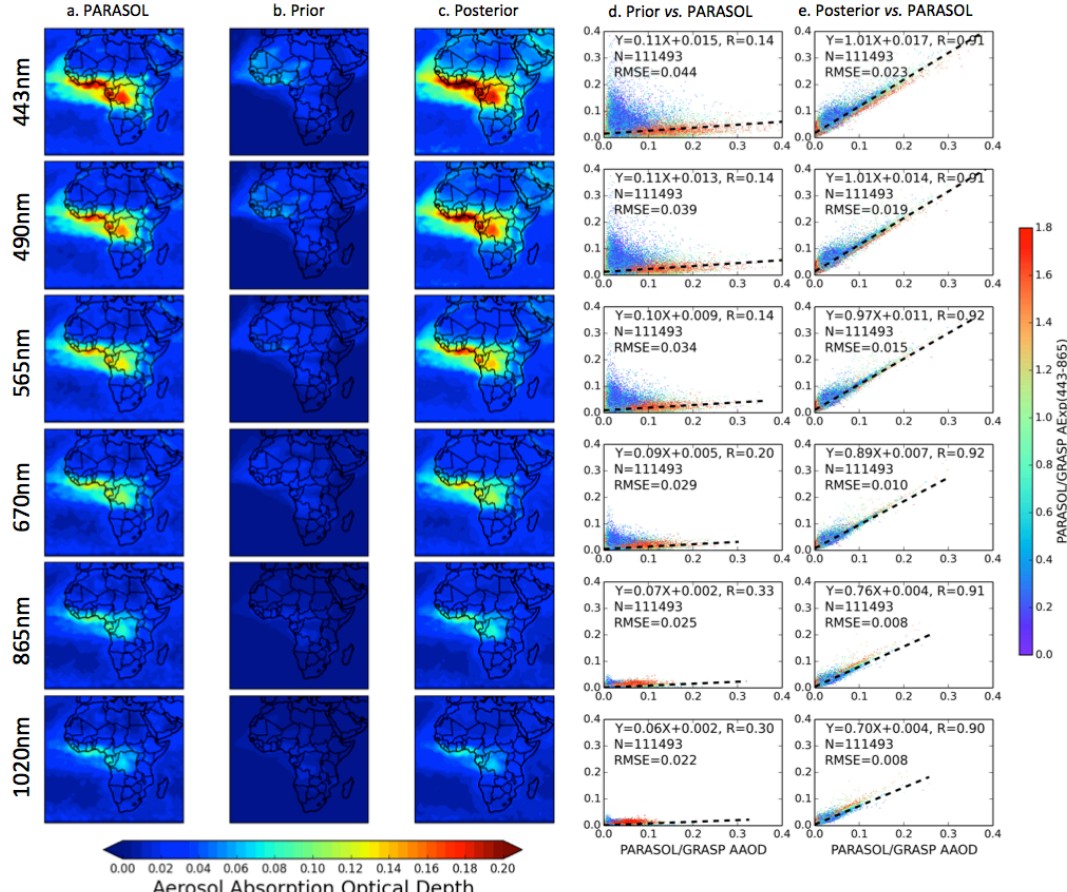

**Figure 12.** Same as Figure 11, but for AAOD.

Figures 12d and 12e show the comparisons of PARASOL/GRASP observed AAOD at 6 wavelengths with the
5   corresponding GEOS-Chem simulated quantities using prior or posterior emissions. The very low linear regression slope
between the model simulated AAOD using prior emissions with observations (less than 0.11 over all six wavelengths)
indicates that the prior simulations significantly underestimate the AAOD. In contrast, model simulations with the posterior
simulations improve the slope to 1.01 at 443 nm and 0.70 at 1020 nm. Similar to the case of AOD, the agreements between
the PARASOL/GRASP AAOD data and the model simulations are much better using the posterior emissions than using the
10   prior emissions, with the correlation coefficients increased from 0.14-0.33 to 0.90-0.92 and the root mean square error





decreased from 0.022-0.044 to 0.008-0.023.

### 4.3 Emission sources

The retrieved and prior monthly total DD, BC and OC emission variations over the study area are shown in Figure 13.

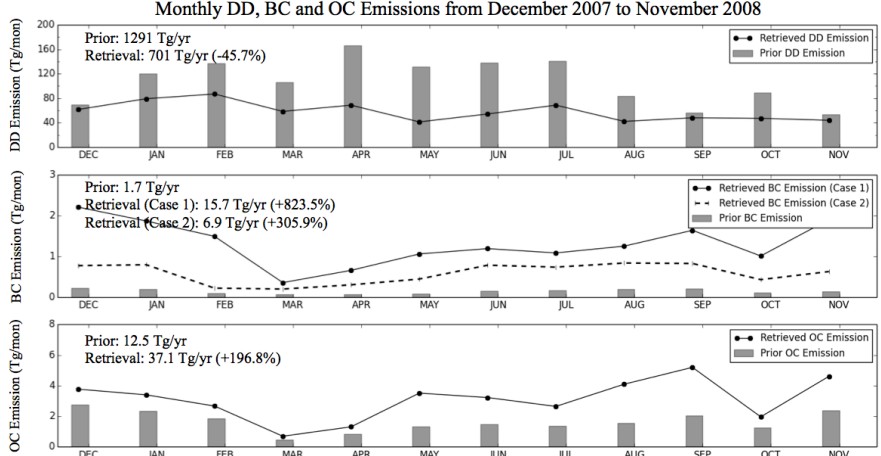

**Figure 13.** Comparison of monthly total DD, BC and OC emissions (unit: Tg mon$^{-1}$) over the study area between prior model (GFED3 and Bond inventories for BC and OC, DEAD model for DD) and retrieved emissions, the annual values (unit: Tg yr$^{-1}$) are provided in the top left corner.

### 4.3.1 DD emissions

Figure 13 shows that the retrieved annual total DD emission in the study area is 701 Tg/yr (particle radius ranging from
0.1 to 6.0 microns, exclude super coarse mode dust particles), which is 45.7% smaller than the prior emissions of 1291 Tg/yr. Moreover, the retrieved total DD emissions show reduced emission amount from the prior values in every month, varying from 11.6% reduction in December to 68.5% in May. Figure 14 shows the comparison of the spatial distribution of seasonal DD emissions between the prior emissions (Figure 14a) and our retrievals (Figure 14b). As shown in Figure 14, the prior and the retrieved emissions show similar spatial and seasonal patterns; for example, the Bodélé Depression is the most active
dust source area in DJF and SON and the Arabian Desert becomes active in MAM and JJA. One major discrepancy between the model and the retrieval is that the model has a much stronger DD sources over Algeria and Morocco in MAM and JJA, which is even stronger than the Bodélé Depression and the Arabian Desert. However, the retrieval still shows the dust emissions there, while the strength reduces a factor of 5-6.


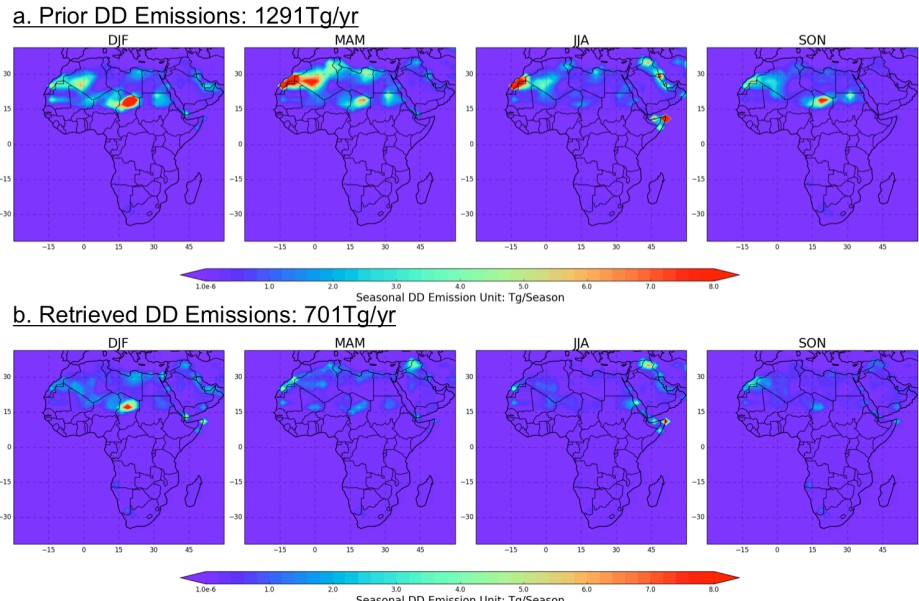

**Figure 14.** Spatial distribution of seasonal desert dust aerosol emission: (a) "prior model" DD emissions from DEAD model and (b) retrieved DD emissions.

### 4.3.2 BC emissions

As mentioned earlier, we considered two cases of BC aerosol refractive index to perform the retrieval (Case 1: $m$=1.75-0.45i; Case 2: $m$=1.95-0.79i), since the retrieved total BC emissions is very sensitive to the BC refractive index (our sensitivity test shows a factor of ~1.8 differences between Case 1 and Case 2, see section 3.2.4). Figure 13 shows the retrievals increase BC emissions for every month from the prior emissions by factors ranging from 5.9 in March to 14.4 in November with an annual averaged increase of a factor of ~8 in Case 1. For Case 2, the retrieved BC emissions have similar

monthly variation as in Case 1 with a smaller magnitude of increase from the prior emissions, from a factor of 3.3 in March to 4.7 in November with an annual averaged increase of ~3.

The spatial comparison of seasonal BC emission is summarized in Figure 15. We plot model prior BC emission from GFED3 and Bond anthropogenic inventories in Figure 15a, retrieved BC emissions from Case 1 in Figure 15b, and Case 2 retrieved BC emissions in Figure 15c. Note that the color bar range in Figure 15b is 2.5 times larger than that of Figure 15a

and Figure 15c. Not surprisingly, the patterns of model prior emission in Case 1 and Case 2 retrievals are similar, with the highest BC emission source areas located in biomass burning regions, such as Central Africa during DJF and Southern Africa JJA. The large increases in the BC emissions in the retrieval relative to the prior suggests that the current model





simulated AAOD is much too low, which is consistent with the PARASOL/GRASP observations in Section 4.2. Retrieval Case 2 shows a large increase over the Arabian Peninsula, indicating there is an emission ~5 times higher than the prior model in DJF, MAM and SON, where the latter shows only a small amount of carbonaceous fine particles. AERONET ground-based measurements indicate a moderate absorption phenomenon there (Seasonal AAOD at 550 nm about ~0.05, see

Figure 21), which corroborates the retrieved values from the inversion.

### 4.3.3 OC emissions

The annual total OC emissions in Figure 13 shows that the retrieved annual OC emissions are higher than the prior model by a factor of ~2, with a minimum monthly increase found in March (1.54) and a maximum in May (5.71). Combined with BC emission, the retrieved total carbonaceous aerosol emissions are 52.8 Tg/yr (with Case 1 BC) and 44.0 Tg/yr (with

Case 2 BC), which is 271.8% (Case 1) to 209.8% (Case 2) higher than prior model (14.2 Tg/yr). We compare the seasonal distribution of prior OC emissions with retrieved emissions in Figure 16. Both the retrieved and prior emissions have highest OC emissions in Southern Africa in JJA and in Central Africa in DJF.

### 4.3.4. Summary of retrieved emissions

Comparison of retrieved DD, BC and OC aerosol emissions over the study area with the GEOS-Chem prior model emission inventories show basically a consistent of spatial and temporal variation. However, the significant differences are in the emission strength. The PARASOL/GRASP based retrieval reduces the GEOS-Chem annual DD emission to 701 Tg/yr over the study area. A recent study by Escribano et al. (2017) estimated that the mineral dust flux for particle size less than 6.0 microns over northern Africa and the Arabian Peninsula is between 630 and 845 Tg/yr. Some other studies also show similar dust emission flux over Africa (Werner et al., 2002; Miller et al., 2004; Escribano et al., 2016, 2017). However, the

overestimation of the prior model dust emission could also result from errors in particle size distribution, which is shown to biased toward smaller particle sizes compared to the observation in the atmosphere (Kok et al., 2017). Meanwhile, the retrieval increases the model annual carbonaceous aerosol emission by about 2.5 times. This value is close to the recommendation given in Bond et al. (2013) that increasing global BC absorption by a factor of 3 to fit the observation of columnar aerosol absorption. In addition, there are many other efforts to improve the simulation of AAOD, e.g. treating

hydrophilic BC as an internal aerosol core with other soluble hygroscopic aerosol species (Wang et al., 2016); including the light absorbing brown carbon in the simulation (Wang et al., 2014b). These studies are all crucial to improve current CTMs aerosol simulation, which should be adopted in our aerosol emission inversion framework in the future.





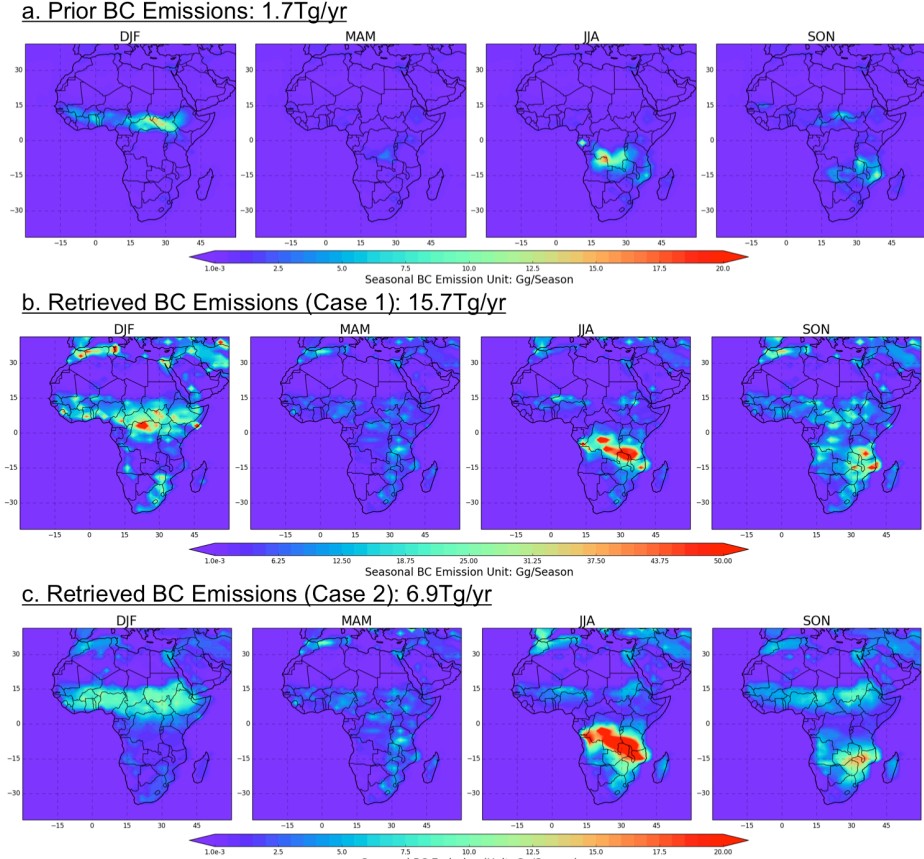

**Figure 15.** Spatial distribution of seasonal BC emissions: (a) prior model BC emissions from GFED3 and Bond inventories; (b) Case 1 retrieved BC emissions; (c) Case 2 retrieved BC emissions. Note that the color scale for (b) is different from (a) and (c) for better resolving the spatial contrasts.



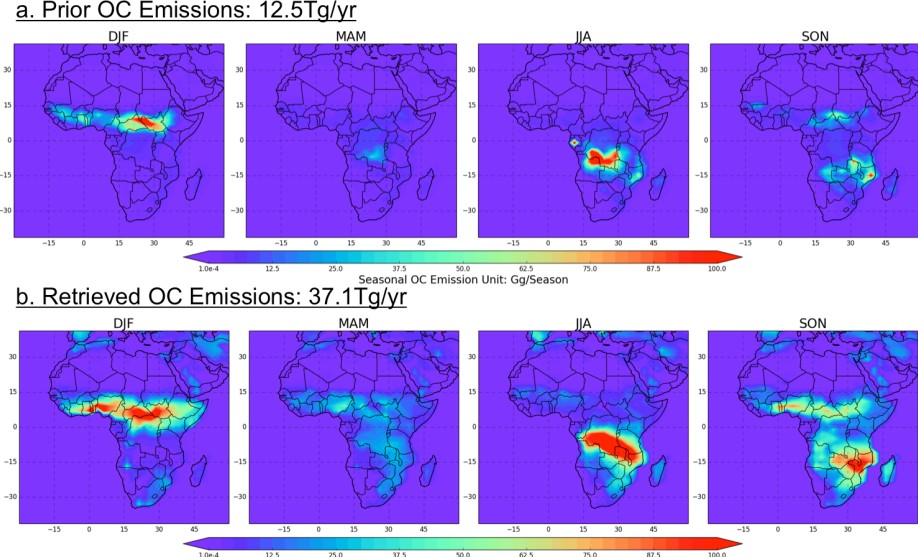

**Figure 16.** Spatial distribution of seasonal OC emissions: (a) prior model OC emissions using GFED3 and Bond inventories and (b) retrieved OC emissions.

## 5 Evaluation

### 5.1 Evaluation with AERONET

In order to objectively evaluate our retrieved aerosol emissions based on PARASOL/GRASP spectral AOD and AAOD, we made a series of evaluations using independent datasets and models not used by our inversion. First, the posterior simulated one-year AOD and AAOD are compared with the sun photometer measured AOD and AAOD at 28 AERONET sites (shown in Figure 1).





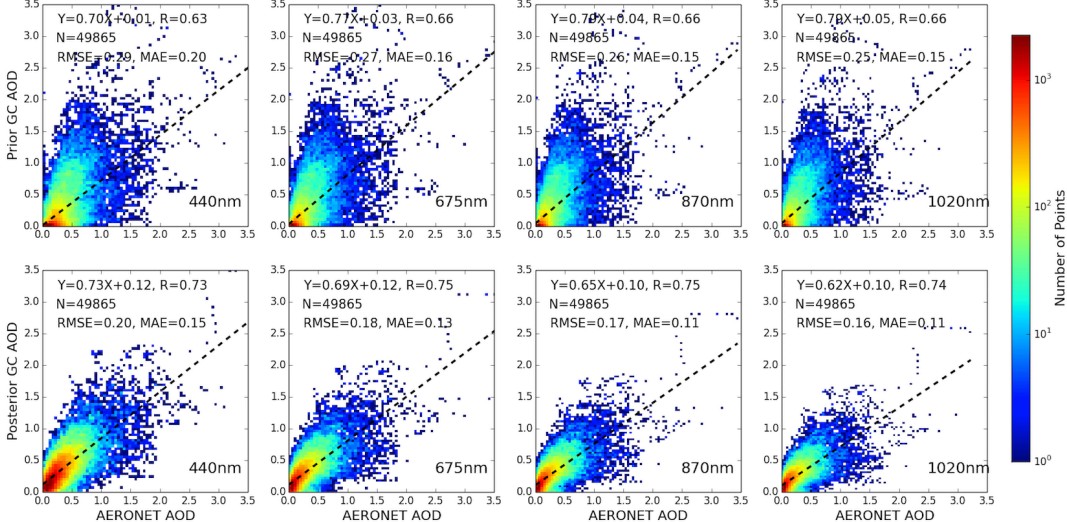

**Figure 17.** Density scatter plots of one-year GEOS-Chem simulated AOD using the prior emissions (top row) or the posterior emissions (bottom row) versus AERONET measured AOD at 440, 675, 870 and 1020 nm at 28 sites. The number of matched pairs (N), correlation coefficient (R), root mean square error (RMSE) and mean absolute error (MAE) are shown on each panel.

Figure 17 and 18 show the comparison of GEOS-Chem simulations using prior and posterior emissions with AERONET measurements of AOD and AAOD, respectively. The evaluation was conducted at 4 wavelengths (440, 675, 870 and 1020 nm) and GEOS-Chem hourly spectral AOD and AAOD are interpolated based on the Ångström exponent. AERONET AOD and AAOD averaged ±30 minutes centered by model output time are used to compare with the model simulations over the grid box containing the AERONET sites. Density scatter plots of 49,865 matched pairs of AOD are shown in Figure 17. The correlation coefficients between GEOS-Chem simulations with prior emissions and AERONET data (shown in upper four panels) are 0.62, 0.67, 0.66 and 0.66 for the four wavelengths respectively, and the corresponding root mean square errors are 0.28, 0.25, 0.24 and 0.24. Yet, the correlation coefficients are increased to 0.73, 0.75, 0.75 and 0.74 when the posterior emissions are used in GEOS-Chem simulation (shown in lower four panels), and meanwhile the root mean square errors are decreased to 0.20, 0.18 0.17 and 0.16, respectively. Meanwhile, the mean absolute errors are also decreased from prior (0.20, 0.16, 0.15 and 0.15) to posterior (0.15, 0.13, 0.11 and 0.11).



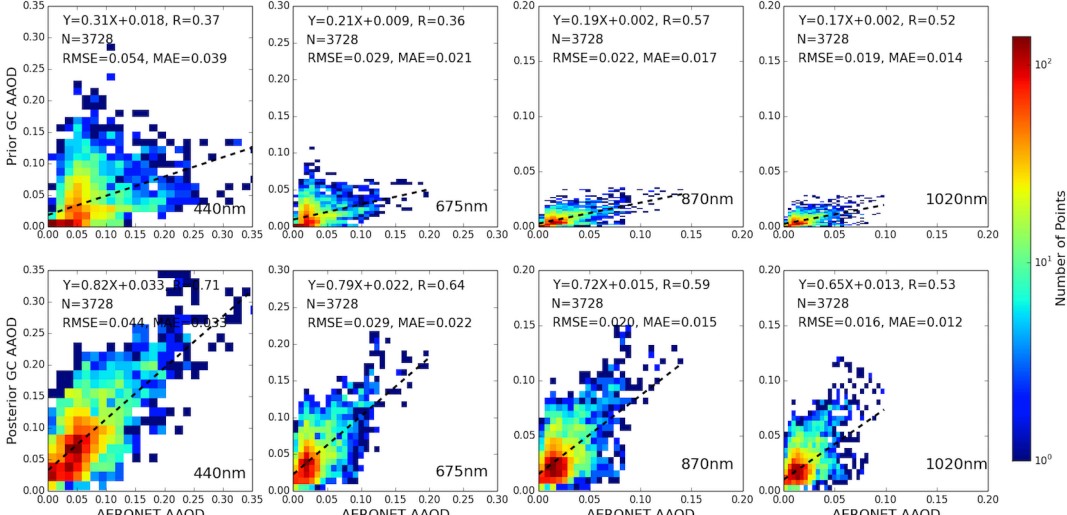

**Figure 18.** Same as Figure 17, but for AAOD.

Figure 18 shows the density scatter plots comparisons for AAOD. However, unlike sun direct measurement of AOD,
AERONET AAOD is inverted from almucantar measurements. To select sufficiently accurate retrievals, we applied standard
quality-screening criteria (e.g. Dubovik et al., (2002b) and Holben et al., (2006)). Therefore, there are fewer AERONET
AAOD that matched with GEOS-Chem simulations than for AOD. The number of matched pairs is 3,728. The low slope of
the linear regression between prior model AAOD and AERONET (shown in upper four panels) indicates that the prior model
significantly underestimates AAOD. The posterior GEOS-Chem simulations using retrieved emissions (shown in the lower
four panels) shows the improvements validating with AERONET, with the correlation coefficients come to 0.71, 0.64, 0.59
and 0.53. In addition, the root mean square errors are also improved for posterior simulations.

Comparison between time series of AOD and AAOD at 440 nm from AERONET, PARASOL/GRASP, and prior and
posterior GEOS-Chem simulations from December 2007 to November 2008 are made in two AERONET sites (Mongu and
Ilorin), and the results are shown in Figure 19. The geo-locations of these two sites are already apparent in Figure 1. Ilorin is
located close to the active dust sources in the Northern Africa, where are also influenced by seasonal biomass burning events,
especially from November to February. Mongu is located close to the Southern Africa seasonal biomass burning sources.
The posterior simulations better capture the time series variations and magnitude of AOD and AAOD from AERONET
measurements. For example, in Mongu, the prior simulation underestimates AOD and AAOD significantly. In September,
the underestimations are about 3 times (a bias of -0.56 for monthly average) for AOD and 4 times for AAOD (a bias -0.09).
Such bias is significantly reduced to -0.22 for AOD and +0.01 for AAOD in posterior simulation with retrieved emissions. In



terms of correlation coefficients, the prior GEOS-Chem simulation shows a solid correlation with measurements in Mongu, while the slope of the linear regression (K) between prior simulation and AERONET (0.24 for AOD; 0.22 for AAOD) indicates that the model significantly underestimates the aerosol loading in Mongu. Furthermore, prior GEOS-Chem simulation can capture the variation and magnitude of AOD (R=0.79 and K=0.79) in Ilorin. However, for AAOD, the

simulation shows underestimation with slope K=0.40, which is an indicator of the model underestimation of the aerosol absorption species, such as BC. Overall, the posterior GEOS-Chem simulation with retrieved emissions can better capture the time serial variation and magnitude of AOD and AAOD in both Mongu and Ilorin.

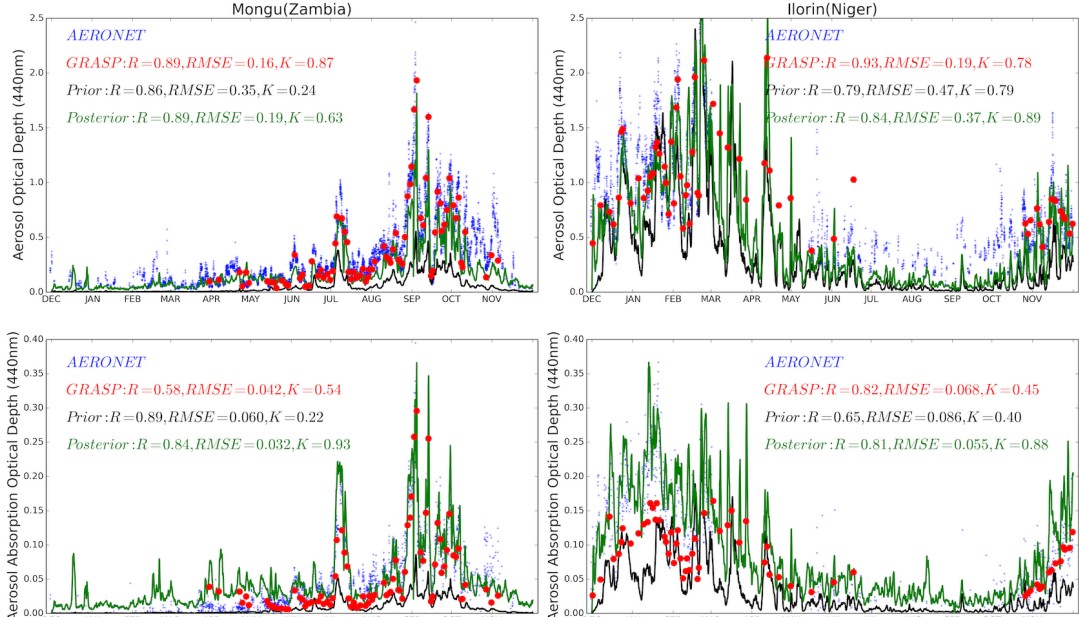

**Figure 19.** Time serial AOD (upper panel) and AAOD (lower panel) from AERONET (blue crosses), PARASOL/GRASP
(red circles), prior GEOS-Chem (black line) and posterior (green line) GEOS-Chem simulations at 2 sites (Mongu and Ilorin). The error statistics parameters between PARASPOL/GRASP, prior and posterior GEOS-Chem simulations with AERONET are also shown in the figure.

### 5.2 Testing retrieved emission in the GEOS-5/GOCART model

All the evaluations considered thus far are a based on simulations in the GEOS-Chem model. To evaluate how such
results may be impacted by model biases owing to factors other than BC, OC and DD emissions, here we ask - can aerosol emissions retrieved from the GEOS-Chem based inversion improve the aerosol simulation for another chemical transport



model? To investigate this, we implement our PARASOL/GRASP based aerosol emission database into the GEOS-5/GOCART model (Chin et al., 2002, 2009, 2014; Colarco et al., 2010). The prior and posterior GEOS-5/GOCART model simulated seasonal AOD are compared with MODIS observations in Figure 20. GEOS-5/GOCART uses similar meteorological fields as GEOS-Chem, with the prior anthropogenic emissions from the Hemispheric Transport of Atmospheric Pollution (HTAP) Phase 2, biomass burning emissions from the Fire Energetics and Emission Research (FEER) database (Ichoku and Ellison, 2014), dust emission calculated as a function of 10-m winds and surface characteristics (Ginoux et al., 2001), and volcanic emissions from OMI-based estimates (Carn et al., 2015). The PARASOL/GRASP retrieved DD, BC, and OC emissions over the study domain are used in the "posterior" simulations while other sources remain unchanged. On an annual average, the DD, BC, and OC posterior/prior emission ratios in the study area are 0.53, 5.3, and 1.2, respectively.

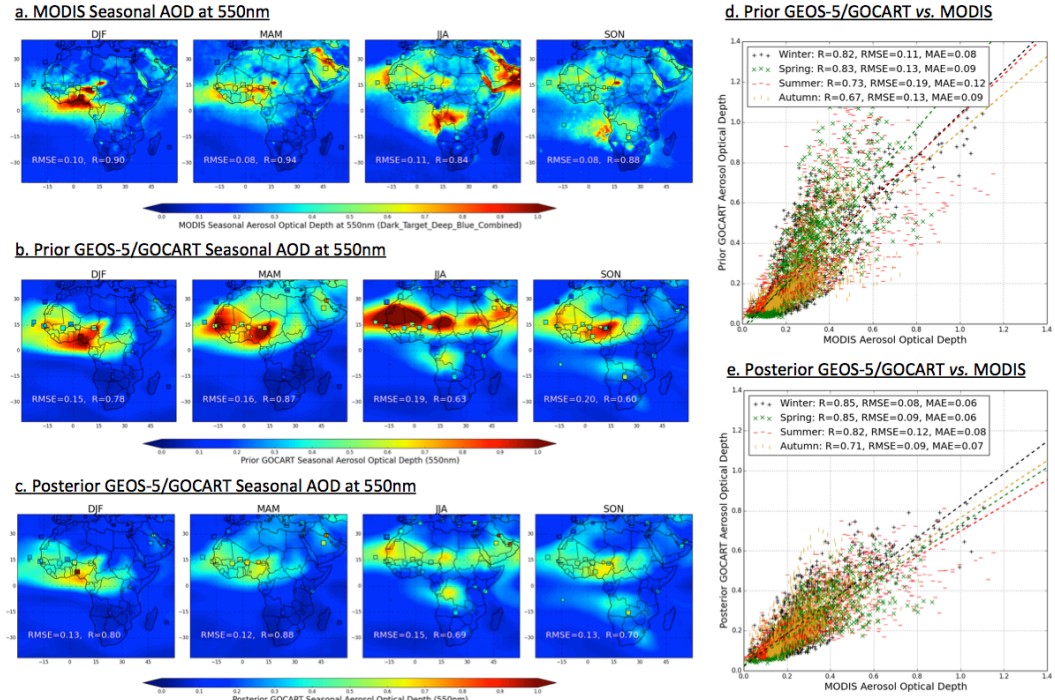

**Figure 20.** Comparison of the seasonal spatial distribution of prior (b) and posterior (c) GEOS-5/GOCART simulated AOD at 550 nm with MODIS observations (a). The scatter plots of grid-to-grid comparison between MODIS and prior GEOS-5/GOCART AOD (d) and posterior GEOS-5/GOCART AOD (e) are also shown. The ground-based measurements from AERONET (squares) are over plotted over figures a-c. The MODIS and GEOS-5/GOCART versus AERONET correlation




coefficient (R) and root mean square error (RMSE) are provided in figures a-c. Meanwhile, the GEOS-5/GOCART versus MODIS R, RMSE and MAE are also provided in figures (d-e).

Figure 20a shows the MODIS seasonal AOD at 550 nm. In order to have better spatial coverage, we take MODIS collection 6 combined dark target and deep blue AOD products at the spatial resolution of 1° x 1° (Hsu et al., 2004; Levy et al., 2013). Figure 20b presents prior GEOS-5/GOCART simulated seasonal AOD, and Figure 20c shows the posterior GEOS-5/GOCART simulation from our retrieved emissions (using Case 2 BC emission). In Figure 20d and 20e, we plot the grid-to-grid comparison between GEOS-5/GOCART prior and GEOS-5/GOCART posterior AOD with MODIS respectively; here the different colors represent different seasons. In order to carry out this grid-to-grid comparison, MODIS 1° x 1° AOD is re-gridded to the resolution 2.0° x 2.5°. The prior GEOS-5/GOCART simulate optical depth is comparable to MODIS observations with similar spatial pattern and correlation coefficient with MODIS R=~0.75 over a year. In addition, the simulation is better in DJF and MAM than in JJA and SON. The correlation coefficient with MODIS is about 0.82 and the root mean square error is about 0.12 in DJF and MAM, and it has a relatively low correlation in JJA and SON (~0.7); meanwhile the RMSE becomes high (~0.16). The prior GEOS-5/GOCART simulation somewhat overestimated observations over the Northern Africa dust region over 4 seasons, while it is underestimated in the southern Africa biomass burning area, especially in biomass burning seasons (JJA and SON), which can also be inferred from the validation with AERONET measurements (squares) over plotted in Figure 20a-c. With the posterior emissions, the GEOS-5/GOCART simulation shows improvements compared with AERONET and MODIS observations, with higher correlation coefficient and lower root mean square error in all 4 seasons than the prior GEOS-5/GOCART simulation. The posterior GEOS-5/GOCART simulated AOD is a little lower than MODIS on average 13% (Normalized Mean Bias, NMB=-13%, NMB=$\sum(M_i - O_i)/\sum O_i$, where sums are over the ensemble of all data $i$, and $M_i$ and $O_i$ are the modeled and observed values), likely associated with that the MODIS AOD is observed at noon, however the GEOS-5/GOCART AOD is average over 24 hours during a day.





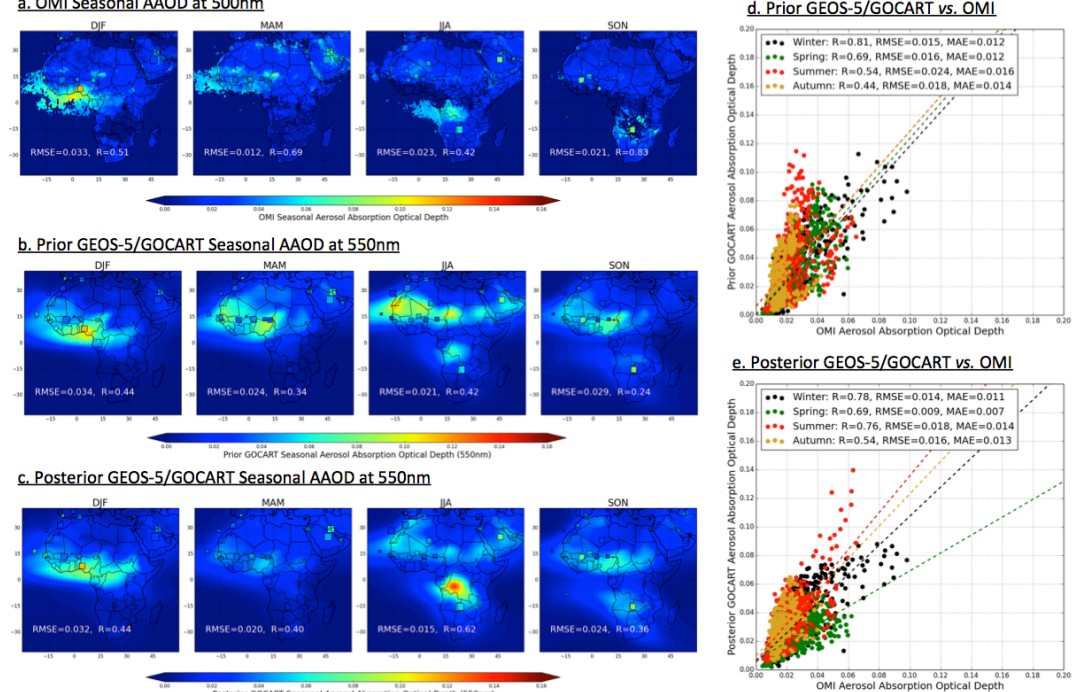

**Figure 21.** Comparison of the seasonal spatial distribution of prior (b) and posterior (c) GEOS-5/GOCART simulated AAOD at 550 nm with OMI observations (a). The scatter plots of grid-to-grid comparison between OMI and prior GEOS-5/GOCART AAOD (d) and posterior GEOS-5/GOCART AAOD (e) are also shown. The ground-based measurements from AERONET (squares) are plotted over figures a-c. The correlation coefficient (R) and root mean square error (RMSE) versus AERONET are provided in figures (a-c). Meanwhile, the GEOS-5/GOCART versus OMI R, RMSE and MAE are also provided in figures (d-e).

Because only ultraviolet and shortwave visible channels and polarimeter measurements are sensitive to aerosol absorption properties, long-term records of AAOD are limited to AERONET, PARASOL/GRASP and OMI. We use the latest OMI aerosol products (OMAERUV version 1.7.4) (Torres et al., 2007, 2013) to evaluate the GEOS-5/GOCART model simulated AAOD from prior aerosol emission inventories and our retrieved aerosol emission database. Meanwhile, collocated AERONET data over the study area are also employed to the evaluation. Detailed assessments of OMI aerosol products are described in other studies (Torres et al., 2013; Ahn et al., 2014; Jethva et al., 2014). Figure 21 shows the validation results. Figure 21a presents the OMI seasonal mean AAOD with original OMAERUV version 1.7.4 spatial





resolution 0.5° x 0.5°. We plot grid-to-grid comparison between OMI and GEOS-5/GOCART AAOD in Figure 21d-e, here OMI AAOD are re-scaled to the same resolution with model simulation 2.0° x 2.5°. Any 2.0° x 2.5° grid box with less than 10 OMI original AAODs (~50% coverage) for averaging is abandoned. This evaluation highlights the following major findings:

1.  The major discrepancy between OMI seasonal AAOD (Figure 21a) and the prior GEOS-5/GOCART simulated AAOD is that the simulated AAOD is higher than OMI values in the Northern Africa dust regions over all seasons, which can be attributed to the overestimation of dust particle absorption (Chin et al., 2009) and/or the total dust emissions. The posterior GEOS-5/GOCART simulated AAOD shows a similar spatial distribution and magnitude as OMI values over dust regions with reduced differences, although the model is still overall higher than OMI
especially over the Southern Africa biomass regions in JJA.

       2.  As shown in Figure 21a, the correlation coefficients of OMI seasonal AAOD with AERONET vary from 0.42 in JJA to 0.83 in SON, meanwhile the root mean square error is smallest in MAM ~0.012 and largest in DJF ~0.033. Preliminary evaluations shows posterior GEOS-5/GOCART simulated seasonal AAOD (Figure 21c) have a slightly better correlation with AERONET than prior GEOS-5/GOCART simulation (Figure 21b) -- the mean correlation
coefficient over entire year improves from ~0.36 to ~0.46 and the mean RMSE decreases from ~0.027 to ~0.023.

       3.  From the scatter plot of GEOS-5/GOCART simulated AAOD versus OMI AAOD in Figure 21d-e, the significant increase of correlation coefficient from prior to posterior simulations occurs in June-July-August (Prior: 0.54; Posterior: 0.76) as well as decreases of RMSE and MAE (Prior: RMSE=0.024, MAE=0.016; Posterior: RMSE=0.018, MAE=0.014), suggesting the reliability of posterior aerosol emission at high biomass burning
aerosol loading season.

## 6 Conclusions

In this study, we designed a method to retrieve BC, OC and DD aerosol emissions simultaneously from satellite observed spectral AOD and AAOD based on the PARASOL/GRASP retrievals and the adjoint of GEOS-Chem chemical transport model. This method uses prior BC, OC and DD emissions as weak constraints in the inversion by initializing the
retrieval with prior emissions added to uniform background values. A series of numerical tests were performed which show this assumption can provide a better fit to observations, meanwhile it allows the retrieval to produce rather good results even if *a priori* knowledge of emissions is poor. Admittedly, the satellite observations are sparse due to several factors, e.g., the clear-sky condition, global coverage orbit cycle. Nevertheless, the PARASOL 6 wavelengths AOD and AAOD from GRASP algorithm are shown to be sufficient to characterize the distribution and magnitude of BC, OC and DD aerosol emissions
simultaneously under the assumption of DD emissions correction constant over 24h and 4 days correction constant carbonaceous aerosol emissions. The inversion test of synthetic PARASOL-like measurements results in about 25.8% uncertainty for daily total DD emission, 5.9% for daily total BC emission and 26.9% for daily total OC emissions. In




addition, it was shown that using two different assumptions for BC refractive index (Case 1: m=1.75-0.45i; Case 2: 1.95-0.79i) could lead to an additional factor of 1.8 differences in total BC emissions.

We evaluated the GRASP retrieved one-year PARASOL spectral AOD and AAOD with AERONET ground-based observations/retrievals at 28 sites across the study area (30°W-60°E, 40°S-40°N). Good agreements were found even using

rescaling of the retrievals to the spatial resolution of 2.0° x 2.5°. Derimian et al. (2016) and Popp et al. (2016) show similar validation results of PARASOL/GRASP with AERONET. Therefore, we used PARASOL/GRASP retrieved spectral AOD and AAOD to optimize BC, OC and DD aerosol emissions in a year (December 2007 to November 2008) over the study area with horizontal resolution of 2.0° x 2.5° in order to match the adjoint GEOS-Chem spatial resolution. The retrieved emissions are publically available at http://csuchencheng.wixsite.com/chencheng/research-blog; this dataset will be available

soon at the GEOS-Chem inventory findings website (http://wiki.seas.harvard.edu/geos-chem/index.php/Inventory_Findings).

Our analysis of the retrieved aerosol emissions indicates that the prior GEOS-Chem model overestimates annual desert dust aerosol emissions by a factor of about 1.8 (with the DEAD scheme) over the study area, similar to other previous modeling studies (Huneeus et al., 2012; Johnson et al., 2012; Ridley et al., 2012, 2016). The retrieved annual BC and OC emissions show a consistent seasonal variation with emission inventories (GFED3 for biomass burning and Bond for

anthropogenic fossil fuel and biofuel combustions). However, we find these BC and OC emissions to have broad underestimations throughout the study area. For example, emissions from the emission inventories for BC are significant lower than our retrieved values up to 823.5% (Case 1) and 305.9% (Case 2), and for OC they are about 196.8% lower. These results are reflected in the model bias of AOD and AAOD from the prior GEOS-Chem simulation, e.g. significant low bias over the biomass burning regions and high bias over the Sahara desert. Underestimation of BC and OC emissions in

chemical transport models have been suggested previously (Sato et al., 2003; Zhang et al., 2015). However, we cannot rule-out the possibility that differences between model and observations could also be attributed to the errors in removal processes and aerosol microphysical properties, in addition to the deficiencies in emissions (Bond et al., 2013). Nevertheless, the fidelity of our results is confirmed by comparison of posterior simulations with measurements from AERONET that are completely independent from and more temporally frequent than PARASOL observations. Specifically, to analyze the

PARASOL/GRASP based aerosol emission database further, we implemented these emissions in the GEOS-5/GOCART model and compared the resulting simulations of AOD and AAOD with independent MODIS and OMI observations. The comparisons show better agreements between model and observations with the posterior GEOS-5/GOCART results (lower biases and higher correlation coefficients) than prior simulations. In the future, we plan to apply our approach globally to longer records of observations to further investigate the inter-annual variability of aerosol emissions on global scales, and to

test our retrieved emission database in other models.

**Acknowledgments**

This work is supported by the Laboratory of Excellence CaPPA – Chemical and Physical Properties of the Atmosphere





– project, which is funded by the French National Research Agency (ANR). We would like to thank the GEOS-Chem and adjoint GEOS-Chem model developers; DKH recognizes support from NASA ACMAP NNX17AF63G. We also thank the entire AERONET team, and especially the principal investigators and site mangers of the 28 AERONET stations that we acquired data from. The authors are also grateful for the MODIS and OMI aerosol team (O. Torres and H. Jethva) for

providing the data used in this investigation, and Huisheng Bian and Tom Kucsera for incorporating the PARASOL/GRASP emissions into the GEOS-5 model and provide the GEOS-5/GOCART simulation results.

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
