# Peer review of "Retrieval of desert dust and carbonaceous aerosol emissions over Africa from POLDER/PARASOL products generated by GRASP algorithm"

_Atmospheric Chemistry and Physics, 2018_

## Referee Comment (RC1) · Anonymous Referee #1 · 23 Mar 2018

This is a solid contribution. My previous comments have been addressed. I recommend the publication of this paper as is.
* * *

---

## Referee Comment (RC2) · Anonymous Referee #2 · 6 Apr 2018

The manuscript by Cheng et al. presents a top-down emission estimates of desert dust, black carbon and organic carbon aerosols over Africa and the Arabian Peninsula. The authors describe the inversion methodology, perform sensitivity tests and apply their tool to the fitting of one year of AOD and AAOD GRASP retrievals from PARASOL measurements. The authors show an improvement in the simulated AOD and AAOD when the outputs of two models are compared against satellite and ground-based observations.

**General comments:**

The manuscript presents interesting and new results that contribute to the research

field. The sensitivity tests are appropriate, the results presented are very good and the implementation of the posterior emissions in a different model confirms the quality of the results. More importantly, this work shows that it is possible to compute top-down estimates of emissions with a relatively high spatial and temporal resolution. The manuscript shows a detailed and complete work, but the quality of the figures and the general presentation of the text overshadow the work.

Despite the fact that English is not my mother tongue, I have found a large number of grammatical errors. Please correct carefully the English language to accomplish with the journal standards and to improve the readability of the manuscript. I have pointed out some of these errors in the minor changes section (only for the first pages of the manuscript).

In general, the figures are low quality for publication. Labels are small and captions are incomplete as they do not explain well the elements in the figures. In most of the maps there is no latitude/longitude labels, etc. Please follow the guidelines from https://www. atmospheric-chemistry-and-physics.net/for_authors/manuscript_preparation.html

The methodology section (3.1) is not clear. The notation is not consistent throughout the text and there are elements in the equations that are not explained. I would suggest to avoid details about the minimization procedure (they are already explained in Dubovik et al 2008) and to focus on the improvements and differences with the Dubovik et al 2008 work.

Basic information about the assimilation is missing in the manuscript. The definitions and values of the error covariance matrices and the regularization parameter are nowhere stated. There is a whole subsection about the weights of the observational error covariance matrix without knowing the error covariance matrix. How do the authors account for the model error in their assimilation? How valid is the diagonal assumption of the observational error covariance matrix, knowing that the assimilated AOD and AAOD are issued from the same algorithm and measurements? Do the GRASP

algorithm reports uncertainty of AOD and AAOD? Are these uncertainties taken into account in this work? Which are the uncertainties of the emissions factors? Are they the same for the different aerosol types, locations and seasons?

Reporting uncertainties of the posterior emissions would be an additional major contribution of this work, but in practice it is not always possible to compute them confidently. Would be possible to provide these values?

**Specific comments:**

Title
Following the ACP guidelines, the title should not be capitalised in every word.

Short summary
I think that the qualifier "important" in "This study is an important contribution to" is out of place (and sounds like a subjective and personal appreciation that do not add information to the summary).

P1 L14-17
Sentence too long. I suggest to split the sentence after "(OC) emissions" and continue with "AOD and AAOD from ... has been assimilated .. ".

P1 L18
These tests show..

P1 L19
Remove "For example"

P1 L21
Please rephrase "...an additional about 1.8 times differences..."

P1 L24-25
Remove "GEOS-Chem inventory of". The inventory is not from GEOS-Chem, it is from Bond et al. and GFED.

P1 L29
Please change "independent of and more"

P1 L31 – P2 L1
I am not sure if it is appropriate to list all the statistics (as acronyms) of the comparison in the abstract.

P2 L6-8
Despite it is cited in the next sentence, this sentence would improve with a reference.

P2 L14
".. the role that atmospheric.. "

P2 L15
Please clarify that it is about short-wave radiation. The measurements are not only due to aerosols, the interaction of light with molecules, surface and clouds is also important.

P2 L24
"aerosols"

P2 L24
To my knowledge, sensors provide measurements, algorithms can provide retrievals.

P2 L27-28
Reference needed. All the cited satellite retrievals of aerosols are based on visible, UV and near infrared measurements, so this sentence do not add new information. I would say that, given the current state of the instruments, algorithms and knowledge of the system, UV and polarimetric measurements are needed to better retrieve absorption properties of aerosols in the visible.

P2 L29
"high degree" sounds odd.

P2 L30

Please change "answering question regarding" for something more specific.

P2 L30
"fates"?

P2 L31
I think that "incorporate" is not the best word

P3 L3
I would add aerosol processes... "of atmospheric and aerosol processes.."

P3 L4
Please change "are found to"

P3 L12
"i.e. fitting satellite observations and model estimates and by adjusting aerosol emissions". This sentence is not clear and it is inaccurate. Inverse modelling is also applied in mathematics, geophysics, etc (there are a lot of examples in the Tarantola (2005) book). Please be more concise in the definition of inverse modelling.

P3 L13
Please avoid these kind of statements. The data is not inverted. Is the CTM (and the emission model/inventory) what is usually called "inverted".

P3 L14
Please specify which kind of distributions: spatial? temporal? size distributions?

P3 L17
The emissions are not from MODIS AOD.

P3 L17-18
Please remove "works such as"

P3 L3-26
This long sentence could be written better.

P3 L26-27
Is not POLDER the name of the instrument? (and PARASOL the satellite/mission)?

P3 L30
"decreases sharply.." This is true with respect to the wavelength. Wavelength is not the only way to account for the spectrum.

P3 L30-31
Please change "most strongly". "ubiquitously" sounds odd in this context.

P3 L31-32
Please decide : "shortwave", "short-wave" or "short wave". It could be possible to remove "shortwave" from here, as it is followed by "visible".

P3 L34
I do not understand the point of this sentence. MODIS and MISR also provide AOD over bright surfaces. May be it is better to show the accuracy of GRASP retrievals over the desert.

P4 L3-6
Should not this sentence go in the model description section?

P4 Section 2.1:
Please check the grammar of this section.

P4 L15-16
The variability has drawn the research? Please rephrase.

P4 L16-20
Sentence seems too long. I would prefer: Figure 1 shows the number of ..., the 28 AERONET sites,.. etc.

P4 L23
"Northern Africa Sahara and Arabian Peninsula desert region"?

P5 L7
Please remove or explain the word "forward".

P5 L7
Please correct "with 47 layers vertical resolution"

P5 L10
"consist of" sounds odd.

P5 L11
Please correct the grammar.

P6 L2-6
Remove parenthesis in the citations of lines 2 and 3; check the grammar of the paragraph ("turbulent mixing of particles to the surface"? and all the line 5)

P6 L7
Check "for all"

P6 L8
".. width for ..".
Where are these parameters specified? Please add reference to the table if needed.

P6 L10 and L20
Hygroscopic, not hydroscopic

P6 L14
Is it really necessary to write "aerosol particle"?. Only with "aerosol" (or particle) should be enough.

P6 L22, figure 2a
Why do the authors show $Q_{\mathrm{ext}}/r$ and not $Q_{\mathrm{ext}}$?

P7 L3
This is a personal blog. Please consider uploading the data as supplementary material.

P7 L10-13
The authors should clarify what "to model fully adequate" means.

Table 1.
Would be possible to add the parameters of the rest of the aerosols simulated? (SS and SU)

P8 L5
I suggest to change "is an efficient tool" to "can be used as a tool"

P8 L20-21
Please rephrase and avoid qualifiers as "highly versatile and accurate" without referring to appropriate validations.

P9 L7-8
This sentence is not clear enough and it should be removed.

P10 L4
Please specify which version of AERONET products. (2, 3?)

P10 - P13 Section 3.1
Please read the related general comment. This section is full of mistakes and imprecisions, so I will list only some of them. The authors should note the $\mathbf{C}_{(\cdot)}^{-1}$ are the inverse of the error covariance matrices and not the error covariance matrices. In this section there are several inconsistencies between the equations, and also in the text (for example, the matrix C of equation 5 is not the same as the one of equation 3. $\gamma_r$ of eq. 3 is $\gamma$ in eq. 4, $\Delta f$ is not explained, etc). It is not clear what is $\mathbf{K}^{\text{obs}}$ (matrix of Jacobians? with respect to which variables? what are the "characteristics"?). The symbol $J_p$ is important but it is not explained. The mass is written as $\mathbf{M}$ in the text and m in the equations. The definition of adjoint of L12-13 (P11) is vague and incomplete. An appropriate definition of adjoint operator can be found in the equation 2 of Talagrand and Courtier (1987).

**P12 L14**
Here the authors state that the minimization is performed with the L-BFGS-B algorithm, but equation 3 described the minimization by using a simpler steepest descend algorithm. Do the authors use the algorithm of equation 3 or do they use L-BFGS-B with the gradient as equation 9?. Please clarify.

**P14 L5**
What is a "well-known qualitative tendency"?

**P14 L10**
The authors should clarify in which scenario (A, B, C, D, E) were these tests performed.

**P14 L18-19**
This statement is in general not true. For example, taken the f function equal the identity in equation 3, the cost function is a quadratic function of $S$ in $R^n$, and the problem is well-constrained (as $J$ is a convex -and smooth- function, the problem has an unique solution), and the solution depends on the prior information.

**P15 L7-9**
Please add an introductory paragraph before L9.

**Figure 7**
Should not be divided by surface area?

**P17 L8-9, L16**
Not using a priori knowledge of aerosol emissions implies much more than the B settings. It is not equivalent.

**P17 23**
The false source generation is prevented only over ocean; but it is still allowed to generate a source in the wrong place and time. Please clarify.

**P17 25**
Why do not the authors show this ratio for the retrievals A and B? , how is the uncertainty of this ratio computed?

P18 , Section 3.2.3
Beyond the synthetic tests, how do the authors account for the (known) sub-daily emission variability of DD, BC and OC emissions?

P20 , Section 3.2.4
Which are the emissions used for the "retrieval E"? I would guess that they are the same as "retrieval C", but it is not clear. Are DU and OC included in this test? Is there any difference in retrieved DD and OC by changing the BC refractive index?

Figure 10:
Are these gridpoint values? Are they the accumulated emissions (16 days) of day-by-day emissions? What is the grey area (20% of what?)?, the lines? What is "Y", "X", "R", etc.?

P21 L8-22
These conclusions about the idealised test are presented as a fact (and largely extrapolated to other contexts) without taking into account the nature of the synthetic measurements, and the limitations of the whole data assimilation system. I would suggest to present them as the authors' choice regarding the parameter configuration of the assimilation procedure.

P21 L11
Why are needed 6 and not, for example, 5 wavelengths? Did the authors try with less wavelengths?

P21 L13
Please check the grammar

P21 L18
More stable and accurate than what? Where do the authors show the "stability" of the retrieval in the text? In which sense it is stable?

P21 L19
Please check the first sentence. It should not be inverted (retrievals sensitive to refractive index) ?

P21 L25
Why are not SS and SU included? Errors in the emission of these aerosols will impact the quality of the posterior emissions. How is this taken into account?

P21 L22 or L24
I would recommend to indicate that this configuration/parameters of the assimilation procedure will be used in Sections 4 and 5.

P21 L28
Even though a fixed number of iterations is a very practical stopping criterion, do the authors compute any diagnostic on the optimality of the cost function after 40 iterations?

P21 Section 4
Please indicate which BC Case (refractive index) is used in the results of Sections 4.1, 4.2, 4.3.1 and 4.3.3.

P24 L13-14
Why could the coarse resolution of the model lead to the spectral differences of AAOD presented here?

P26 Section 4.3.1
How much of the DD reported in Figure 13 is produced in the Sahara? The retrieved emission seasonal cycle seems flat (but maybe it is only the scale of the plot). In comparison with other studies, would the authors think that the seasonal cycle is well captured?

Figure 13.
The units are Tg/month or Tg?

P27 L11
This value is similar to the 3.4 factor of Kaiser et al 2012. I suggest to add and comment this reference in line 22.

The authors have indicated that the difference between Case 1 and Case 2 retrievals is close to 1.8 for BC. This value have been computed in the sensitivity tests; but in this section the factor should be close to 8/3 ( 2.7). How do the authors explain this difference? Should not be better to include in the abstract and conclusion this value instead of the 1.8?

P27 L16-20
Please note that these values are not necessarily for the year 2008.

P27 L27
Can the authors report the uncertainty of the retrieved emissions?

P30 L5
Please indicate which BC case is used in this section.

Figure 18
The colour code of each square in the panels represent the number of pairs (observation,model) that fit in the square. The size of the squares are different in all the panels (in fact, some of them are not squares, despite the same limits of the x and y axes), so they are not comparable. The worst case is in the last column. The authors should write in the caption the size of the AAOD bins (and AOD for Figure 17). I strongly recommend to improve this figure.

P35 L22
This sentence should be written before (around line 10). Is the model sampled according to MODIS availability?

P36 L10-11
I could understand that these kind of measurements are more sensitive to the absorption properties, thus the retrieval of them is better constrained, but this first sentence of the paragraph implies that these are the only measurements sensitive to the absorption, which is a strong statement. Could you provide a reference on this?

P36 and P37
From panels a, b and c of Figure 21, it seems that the model/satellite comparison is not collocated. This could introduce errors in the analysis of the results, and it should be mentioned. Also, I would suggest to plot with transparent colour the missing data, and not with blue (which is equivalent to zero AAOD)

P38 L17
Please rephrase this sentence to improve readability (It is not easy to figure out what a "800% lower" means).

**References:**

Talagrand, O. and Courtier, P. Variational assimilation of meteorological observations with the adjoint vorticity equation. I: Theory. Q. J. R. Meteorol. Soc. 113,1311–1328, 1987.
Kaiser, J. W., et al. Biomass burning emissions estimated with a global fire assimilation system based on observed fire radiative power, Biogeosciences, 9, 527–554, 2012.
Tarantola, A. Inverse problem theory and methods for model parameter estimation. siam, 2005.

---

## Author Comment (AC1) · 11 Jun 2018

**Response to comments on "Retrieval of desert dust and carbonaceous aerosol emissions over Africa from POLDER/PARASOL products generated by GRASP algorithm" by Cheng Chen et al.**

We would like to thank the two referees for their time reviewing the manuscript, and for the helpful feedback provided. Their comments allow us to improve the manuscript by better emphasizing its strength and are of important help to our future research. We have taken them into full consideration and made changes accordingly which we hope satisfying the reviewers. Their comments are repeated below along with our responses (in blue).

**Response to comments by Referee #1**

This is a solid contribution. My previous comments have been addressed. I recommend the publication of this paper as is.

**Response**: We appreciate the referee very much for these comments.

**Response to comments by Referee #2**

GENERAL COMMENTS:

The manuscript presents interesting and new results that contribute to the research field. The sensitivity tests are appropriate, the results presented are very good and the implementation of the posterior emissions in a different model confirms the quality of the results. More importantly, this work shows that it is possible to compute topdown estimates of emissions with a relatively high spatial and temporal resolution. The manuscript shows a detailed and complete work, but the quality of the figures and the general presentation of the text overshadow the work.

Despite the fact that English is not my mother tongue, I have found a large number of grammatical errors. Please correct carefully the English language to accomplish with the journal standards and to improve the readability of the manuscript. I have pointed out some of these errors in the minor changes section (only for the first pages of the manuscript).

In general, the figures are low quality for publication. Labels are small and captions are incomplete as they do not explain well the elements in the figures. In most of the maps there is no latitude/longitude labels, etc. Please follow the guidelines from https://www.atmospheric-chemistry-and-physics.net/for_authors/manuscript_preparation.html

**Response**: We thank the referee for detailed reviews and very helpful comments and remarks. We appreciate the referee's feedback and their recognition of the value that the study offers the scientific community. We will address their concerns to improve the precision, clarity and discussion of the manuscript.

The methodology section (3.1) is not clear. The notation is not consistent throughout the text and there are elements in the equations that are not explained. I would suggest to avoid details about the minimization procedure (they are already explained in Dubovik et al 2008) and to focus on the

improvements and differences with the Dubovik et al 2008 work.

**Response**: Indeed, this work is a continuation of the previous work by Dubovik et al. (2008). The results and community reaction on 2008 paper inspired the authors to continue the effort. This is why, in the text of this paper we tried to show clear link between 2008 and current work. From review comments we realized some deficiencies of our description. The section 3.1 was verified and corrected. The left only equations necessary to understand the changes what were implemented in GEOS-CHEM adjoin for this work: the adjoint operation for AOD and AAOD.

Basic information about the assimilation is missing in the manuscript.

**Response**: Section 3.1 provides the conceptual equations introducing the assimilation. We have somewhat revised the text to link the current work with assimilations overall assimilation activities. However, we think adding more introductory material would not correspond to the character of paper and we refer the readers to other more appropriate introductory papers.

The definitions and values of the error covariance matrices and the regularization parameter are nowhere stated. There is a whole subsection about the weights of the observational error covariance matrix without knowing the error covariance matrix. How do the authors account for the model error in their assimilation? How valid is the diagonal assumption of the observational error covariance matrix, knowing that the assimilated AOD and AAOD are issued from the same algorithm and measurements? Do the GRASP algorithm reports uncertainty of AOD and AAOD? Are these uncertainties taken into account in this work? Which are the uncertainties of the emissions factors? Are they the same for the different aerosol types, locations and seasons?

**Response**: We agree that the manuscript is not clear about definition of covariance matrices of measurements and contribution of *a priori* term. We have added some clarifications. Specifically, we recognize that at present covariance matrices of AOD and AAOD are not available and we use very simple assumptions about covariance matrices. We assume that covariance $\mathbf{C}_{obs}$ matrix is diagonal and we introduce the relative weights for AOD and AAOD. The absolute values are not of importance since minimization procedure, in principle, does not require knowledge of the cost function absolute value.

Also, we used only very small contribution of *a priori* term (almost non), and we reported the value of regularization parameters. This is another weakness of the current study that is planned to be investigated in future efforts.

Reporting uncertainties of the posterior emissions would be an additional major contribution of this work, but in practice it is not always possible to compute them confidently. Would be possible to provide these values?

**Response**: We stated in the manuscript that we can't calculate uncertainties of retrieved emissions due to number of challenges. At the same time, we expected that conducted sensitivity tests provide some information how accurate retrieval can be expected. Therefore, based on the test results we make some estimation of the uncertainties.

SPECIFIC COMMENTS:

Title

Following the ACP guidelines, the title should not be capitalised in every word.

**Response**: Thanks for your suggestion. We revised the title as "Retrieval of desert dust and carbonaceous aerosol emissions over Africa from POLDER/PARASOL products generated by GRASP algorithm".

Short summary

I think that the qualifier "important" in "This study is an important contribution to" is out of place (and sounds like a subjective and personal appreciation that do not add information to the summary).

**Response**: Good suggestion. We revised the short summary as "This paper introduces a method to use satellite observed spectral AOD and AAOD to derive three types of aerosol emission sources simultaneously based on inverse modeling in a high spatial and temporal resolution. This study shows it is possible to estimate aerosol emissions and improve the atmospheric aerosol simulation using detailed aerosol optical and microphysical information from satellite observations."

P1 L14-17

Sentence too long. I suggest to split the sentence after "(OC) emissions" and continue with "AOD and AAOD from ... has been assimilated .. ".

**Response**: We revised this sentence as "In this paper, we use the GEOS-Chem based inverse modelling framework for retrieving desert dust (DD), black carbon (BC) and organic carbon (OC) aerosol emissions simultaneously. Aerosol Optical Depth (AOD) and Aerosol Absorption Optical Depth (AAOD) retrieved from the multi-angular and polarimetric POLDER/PARASOL measurements generated by the GRASP algorithm (hereafter PARASOL/GRASP) have been assimilated."

P1 L18

These tests show..

**Response**: Done.

P1 L19

Remove "For example"

**Response**: Done.

P1 L21

Please rephrase "...an additional about 1.8 times differences..."

**Response**: Thanks for your suggestion. It has been changed to "an additional factor of 1.8 differences".

P1 L24-25

Remove "GEOS-Chem inventory of". The inventory is not from GEOS-Chem, it is from Bond et al.

and GFED.

**Response**: Done.

P1 L29

Please change "independent of and more"

**Response**: Thanks for your suggestion. It has been changed to "that are completely independent measurements and more temporally frequent than PARASOL observations."

P1 L31 – P2 L1

I am not sure if it is appropriate to list all the statistics (as acronyms) of the comparison in the abstract.

**Response**: Agreed. We have removed the statistics of the comparison in the abstract.

P2 L6-8

Despite it is cited in the next sentence, this sentence would improve with a reference.

**Response**: Thanks for your suggestion. We include a reference (Bond et al., 2013) to cite this introduction sentence.

P2 L14

".. the role that atmospheric.. "

**Response**: Done.

P2 L15

Please clarify that it is about short-wave radiation. The measurements are not only due to aerosols, the interaction of light with molecules, surface and clouds is also important.

**Response**: Thanks for your suggestion. It has been corrected to "global mean direct shortwave radiative forcing".

We are not completely sure what the reviewer means by sentence "The measurements are not only due to aerosols, the interaction of light with molecules, surface and clouds is also important". Our discussion was about high uncertainly of radiative forcing estimation due to aerosol. We didn't find any contradiction of our statements with the suggestion of the reviewer.

P2 L24

"aerosols"

**Response**: Done.

P2 L24

To my knowledge, sensors provide measurements, algorithms can provide retrievals.

**Response**: Thanks for your suggestion. It has been changed to "only a few satellite aerosol products can provide retrievals of AAOD".

P2 L27-28

Reference needed. All the cited satellite retrievals of aerosols are based on visible, UV and near infrared measurements, so this sentence do not add new information. I would say that, given the current state of the instruments, algorithms and knowledge of the system, UV and polarimetric measurements are needed to better retrieve absorption properties of aerosols in the visible.

**Response**: Thanks for your suggestion. The sentence has been revised.

P2 L29

"high degree" sounds odd.

**Response**: Thanks for your suggestion. This sentence has been revised as "Despite their ability to provide global coverage in high spatial resolution, …"

P2 L30

Please change "answering question regarding" for something more specific.

**Response**: Thanks for your suggestion. It has been changed to "addressing question regarding".

P2 L30

"fates"?

**Response**: We think the word is used correctly.

P2 L31

I think that "incorporate" is not the best word

**Response**: Thanks for your suggestion. It has been changed to "rely on".

P3 L3

I would add aerosol processes... "of atmospheric and aerosol processes.."

**Response**: Done.

P3 L4

Please change "are found to"

**Response**: Thanks for your suggestion. It has been changed to "are expected to".

P3 L12

"i.e. fitting satellite observations and model estimates and by adjusting aerosol emissions". This sentence is not clear and it is inaccurate. Inverse modelling is also applied in mathematics, geophysics, etc (there are a lot of examples in the Tarantola (2005) book). Please be more concise in the definition of inverse modelling.

**Response**: Indeed, the "inverse modeling" is not well-defined term and allows various interpretations. In our understanding "inverse modeling" is related with assimilation effort where CTM (or similar models) fits observation. Here we follow interpretation by Bennett, (2002). We do not think that

"inverse modeling" should be considered as an equivalent to the term "inversion" in general. For example, Tarantola (2005) book does not use "inverse modeling" term.

P3 L13

Please avoid these kind of statements. The data is not inverted. Is the CTM (and the emission model/inventory) what is usually called "inverted".

**Response**: Thanks for your suggestion. This sentence has been revised as "For example, Dubovik et al. (2008) developed an algorithm for inverting CTM and implemented the approach to retrieve distributions of aerosol emissions using MODIS data."

P3 L14

Please specify which kind of distributions: spatial? temporal? size distributions?

**Response**: Done.

P3 L17

The emissions are not from MODIS AOD.

**Response**: Good suggestion. It has been changed to "Huneeus et al. (2012, 2013) optimize global aerosol emissions using MODIS AOD with a simplified aerosol model (Huneeus et al., 2009)."

P3 L17-18

Please remove "works such as"

**Response**: Done.

P3 L3-26

This long sentence could be written better.

**Response**: Corrected.

P3 L26-27

Is not POLDER the name of the instrument? (and PARASOL the satellite/mission)?

**Response**: Corrected.

P3 L30

"decreases sharply.." This is true with respect to the wavelength. Wavelength is not the only way to account for the spectrum.

**Response**: Yes, we agree. By this sentence we discuss only spectral variability.

P3 L30-31

Please change "most strongly". "ubiquitously" sounds odd in this context.

**Response**: Revised.

P3 L31-32

Please decide : "shortwave", "short-wave" or "short wave". It could be possible to remove "shortwave" from here, as it is followed by "visible".

**Response**: Corrected.

P3 L34

I do not understand the point of this sentence. MODIS and MISR also provide AOD over bright surfaces. May be it is better to show the accuracy of GRASP retrievals over the desert.

**Response**: Actually, it is known problem and the first MODIS "Dark Target" algorithm did not provide any retrieval over bright surfaces. More recent algorithms, such as "Deep Blue", provide retrieval over bright surface, however the accuracy of the retrieval is still limited. For example, Ångström exponent is uncertain and certainly SSA retrieval from MODIS.

This sentence has been revised as "The GRASP retrieval overcomes the difficulty of deriving aerosol over bright surfaces in the visible wavelengths and GRASP provides both AOD and AAOD even over desert that should help improve constraints of DD emissions over source regions, rather than having to rely on downwind observations (e.g., Wang et al., 2012)."

P4 L3-6

Should not this sentence go in the model description section?

**Response**: We believe that removing this sentence will reduce the clarity in explanation of the work done.

P4 Section 2.1:

Please check the grammar of this section.

**Response**: Done.

P4 L15-16

The variability has drawn the research? Please rephrase.

**Response**: Revised.

P4 L16-20

Sentence seems too long. I would prefer: Figure 1 shows the number of ..., the 28 AERONET sites,.. etc.

**Response**: Done.

P4 L23

"Northern Africa Sahara and Arabian Peninsula desert region"?

**Response**: Done.

P5 L7

Please remove or explain the word "forward".

**Response**: Done.

P5 L7

Please correct "with 47 layers vertical resolution"

**Response**: Thanks for your suggestion. It has been changed to "with 47 vertical layers".

P5 L10

"consist of" sounds odd.

**Response**: Revised. This sentence has been revised as "Dust simulations in GEOS-Chem (Fairlie et al., 2007) combine the mineral dust entrainment and deposition (DEAD) model (Zender et al., 2003) with the GOCART dust source function (Ginoux et al., 2001)."

P5 L11

Please correct the grammar.

**Response**: Done.

P6 L2-6

Remove parenthesis in the citations of lines 2 and 3; check the grammar of the paragraph ("turbulent mixing of particles to the surface"? and all the line 5)

**Response**: Done.

P6 L7

Check "for all"

**Response**: Done.

P6 L8

".. width for ..". Where are these parameters specified? Please add reference to the table if needed.

**Response**: Thanks for your suggestion. This sentence has been revised as "The modal ($r_{\text{mean}}$) and effective ($r_{\text{eff}}$) radius, and width (sigm) for each dry aerosol species and their optical properties is specified (see in Table 1)."

P6 L10 and L20

Hygroscopic, not hydroscopic

**Response**: Corrected.

P6 L14

Is it really necessary to write "aerosol particle"?. Only with "aerosol" (or particle) should be enough.

**Response**: Revised.

P6 L22, figure 2a

Why do the authors show Qext/r and not Qext?

**Response**: $Q_{ext}(\lambda)/r_e$ is the extinction efficiency normalized by particle effective radius, which indicates the particle extinction ability in consideration of particle size distribution assumption.

P7 L3

This is a personal blog. Please consider uploading the data as supplementary material.

**Response**: Thanks for your suggestion. We have removed this personal blog in the text and added the "Data availability" section at the end of the text.

P7 L10-13

The authors should clarify what "to model fully adequate" means.

**Response**: Revised.

Table 1.

Would be possible to add the parameters of the rest of the aerosols simulated? (SS and SU)

**Response**: Good suggestion. We have added the parameters of SS and SU in Table 1.

P8 L5

I suggest to change "is an efficient tool" to "can be used as a tool"

**Response**: Revised.

P8 L20-21

Please rephrase and avoid qualifiers as "highly versatile and accurate" without referring to appropriate validations.

**Response**: We agree. This sentence has been revised as "GRASP is a recently developed aerosol retrieval algorithm that processes properties of aerosol and land surface reflectance."

P9 L7-8

This sentence is not clear enough and it should be removed.

**Response**: Done.

P10 L4

Please specify which version of AERONET products. (2, 3?)

**Response**: We agree. AERONET Version 2 data are used in this study, and we have added this information in this sentence.

P10 - P13 Section 3.1

Please read the related general comment. This section is full of mistakes and imprecisions, so I will list

only some of them. The authors should note the $\mathbf{C}_{(\cdot)}^{-1}$ are the inverse of the error covariance matrices and not the error covariance matrices. In this section there are several inconsistencies between the equations, and also in the text (for example, the matrix C of equation 5 is not the same as the one of equation 3. $\gamma_r$ of eq. 3 is $\gamma$ in eq. 4, $\Delta f$ is not explained, etc). It is not clear what is K$^{obs}$ (matrix of Jacobians? with respect to which variables? what are the "characteristics"?). The symbol $J_p$ is important but it is not explained. The mass is written as M in the text and m in the equations. The definition of adjoint of L12-13 (P11) is vague and incomplete. An appropriate definition of adjoint operator can be found in the equation 2 of Talagrand and Courtier (1987).

**Response**: Section 3.1 has been revised. Though, we would like to note that we fully agree that Talagrand and Courtier give more rigorous definition. This paper is not aimed to provide fundamental discussion of this matter.

P12 L14

Here the authors state that the minimization is performed with the L-BFGS-B algorithm, but equation 3 described the minimization by using a simpler steepest descend algorithm. Do the authors use the algorithm of equation 3 or do they use L-BFGS-B with the gradient as equation 9?. Please clarify.

**Response**: We used GEOS-Chem, i.e. L-BFGS-B was used. We added clarifications in the text.

P14 L5

What is a "well-known qualitative tendency"?

**Response**: Revised.

P14 L10

The authors should clarify in which scenario (A, B, C, D, E) were these tests performed.

**Response**: Thanks for your suggestion. In this test, scenario C is used. We have added this information in the text. "The retrievals are conducted with option A and option B respectively (the inversion is conducted under Retrieval C scenario, see in the following sections), with other settings held constant."

P14 L18-19

This statement is in general not true. For example, taken the f function equal the identity in equation 3, the cost function is a quadratic function of S in Rn, and the problem is well-constrained (as J is a convex -and smooth- function, the problem has an unique solution), and the solution depends on the prior information.

**Response**: The text is improved to clarify our statement.

P15 L7-9

Please add an introductory paragraph before L9.

**Response**: Done.

Figure 7

Should not be divided by surface area?

**Response**: The "true" emission is defined at each grid box without consideration of the surface area of each grid box. So, we keep the amount of emission per grid box in Figure 7.

P17 L8-9, L16

Not using a priori knowledge of aerosol emissions implies much more than the B settings. It is not equivalent.

**Response**: We agree, and it has been revised.

P17 23

The false source generation is prevented only over ocean; but it is still allowed to generate a source in the wrong place and time. Please clarify.

**Response**: Revised.

P17 25

Why do not the authors show this ratio for the retrievals A and B? , how is the uncertainty of this ratio computed?

**Response**: This ratio, which is a mean of the ratios between retrieved emission $S_{\text{retrieval}}$ and "true" emission $S_{\text{true}}$ at all grid boxes over study area, can be used to evaluate the performance of the retrieval. However, Retrieval A and B show some limitations to capture the correct spatial distribution of emissions. So, there are some invalidate values to make this ratio for Retrieval A and B (e.g. $S_{\text{retrieval}} > 0.0$ and $S_{\text{true}} = 0.0$). The ratios of them would make no sense. And the uncertainty indicates the standard deviation of the mean ratio.

P18 , Section 3.2.3

Beyond the synthetic tests, how do the authors account for the (known) sub-daily emission variability of DD, BC and OC emissions?

**Response**: In the GEOS-Chem simulation, we adopt daily GFED3 biomass burning emission and monthly BOND inventory. So, the sub-daily BC and OC emission is constant. In the grid boxes, where the prior DD emissions are greater than zero ($S_{0,DD} > 0.0$), we keep the sub-daily DD emission variability as prior DEAD dust model. However, the added background emissions over land grid boxes are sub-daily constant. In the inversion, we correct the DD emission using one scaling factor during a day.

P20 , Section 3.2.4

Which are the emissions used for the "retrieval E"? I would guess that they are the same as "retrieval C", but it is not clear. Are DU and OC included in this test? Is there any difference in retrieved DD and OC by changing the BC refractive index?

**Response**: Yes, the other settings for "Retrieval E" are the same with "Retrieval C" expect for BC refractive index. We have added this information in the text.

In the Retrieval E, we fix the same DD and OC emissions as that of "Retrieval C". We didn't retrieve DD and OC emissions in this test.

Figure 10:

Are these gridpoint values? Are they the accumulated emissions (16 days) of day-by day emissions? What is the grey area (20% of what?)?, the lines? What is "Y", "X", "R", etc.?

**Response**: Yes. This figure shows the grid-to-grid comparison of retrieved emissions (Retrieval C and Retrieval E) with the "true" BC emissions. The values are averaged over 16 days. And the grey area represents ±20% differences around the true values. "Y=kX" is the linear regression statistics between retrieved emissions (y-axis) with the "true" values (x-axis), and "R" is the correlation coefficient. We have added description to Figure 10 caption:

"Figure 10. Test of BC particle refractive index influence on the retrieval of BC emissions. The scatter plots are grid-to-grid comparison of retrieved 16 days averaged emissions (blue: Retrieval C; green: Retrieval E) with the "true" BC emissions. The shade grey area represents ±20% differences around the true values."

P21 L8-22

These conclusions about the idealised test are presented as a fact (and largely extrapolated to other contexts) without taking into account the nature of the synthetic measurements, and the limitations of the whole data assimilation system. I would suggest to present them as the authors' choice regarding the parameter configuration of the assimilation procedure.

**Response**: Done.

P21 L11

Why are needed 6 and not, for example, 5 wavelengths? Did the authors try with less wavelengths?

**Response**: we used maximum possible data, since reduction of the data hardly can bring serious advantages.

P21 L13

Please check the grammar

**Response**: Revised.

P21 L18

More stable and accurate than what? Where do the authors show the "stability" of the retrieval in the text? In which sense it is stable?

**Response**: Revised.

P21 L19

Please check the first sentence. It should not be inverted (retrievals sensitive to refractive index) ?

**Response**: Revised.

P21 L25

Why are not SS and SU included? Errors in the emission of these aerosols will impact the quality of the posterior emissions. How is this taken into account?

**Response**: We do not think that PARASOL data have sufficient information, however we will revisit this conclusion in future studies.

P21 L22 or L24

I would recommend to indicate that this configuration/parameters of the assimilation procedure will be used in Sections 4 and 5.

**Response**: Revised.

P21 L28

Even though a fixed number of iterations is a very practical stopping criterion, do the authors compute any diagnostic on the optimality of the cost function after 40 iterations?

**Response**: Revised.

P21 Section 4

Please indicate which BC Case (refractive index) is used in the results of Sections 4.1, 4.2, 4.3.1 and 4.3.3.

**Response**: Good suggestion. Case 1 BC refractive index is used in the results of this Section. We have added this information in the text.

P24 L13-14

Why could the coarse resolution of the model lead to the spectral differences of AAOD presented here?

**Response**: Because the PARASOL/GRASP retrieval of AAOD is at PARASOL original 6x7 km pixel, and then the PARASOL AAODs are aggregated into model 2.0° x 2.5° grid boxes. There could be different coverage of observations in different grid boxes.

P26 Section 4.3.1

How much of the DD reported in Figure 13 is produced in the Sahara? The retrieved emission seasonal cycle seems flat (but maybe it is only the scale of the plot). In comparison with other studies, would the authors think that the seasonal cycle is well captured?

**Response**: Our annual dust emission is 701 Tg over Africa and the Arabian Peninsula, which is agreed with the recent estimation by Escribano et al. (2017) that between 630 and 845 Tg over this area using observations from MODIS, MISR and PARASOL. Figure S1 shows the seasonal cycle of retrieved dust emission over Africa in comparison with that from prior GEOS-Chem model.

[Figure]

Figure S1. Comparison of monthly DD emission (unit: Tg) over Africa between prior model and retrieved emissions.

Figure 13.

The units are Tg/month or Tg?

**Response**: Corrected. It should be Tg.

P27 L11

This value is similar to the 3.4 factor of Kaiser et al 2012. I suggest to add and comment this reference in line 22. The authors have indicated that the difference between Case 1 and Case 2 retrievals is close to 1.8 for BC. This value have been computed in the sensitivity tests; but in this section the factor should be close to 8/3 ( 2.7). How do the authors explain this difference? Should not be better to include in the abstract and conclusion this value instead of the 1.8?

**Response**: Thanks for your suggestion. We have added Kaiser's 2012 paper on the discussion section 4.3.4. "Kaiser et al. (2012) also recommend correcting the carbonaceous aerosol emission (GFED3) with a factor 3.4 when using them in the global aerosol forecasting system."

Yes, the ratio of Case 1 and Case 2 BC emission from one-year real data inversion is ~2.3 (15.7/6.9), which is different from this ratio 1.8 in sensitivity test than conducted using 16 days synthetic data. This ratio is related to the intensity and spatial distribution of BC emission. Comprehensive tests should be done to analysis it in the future studies.

P27 L16-20

Please note that these values are not necessarily for the year 2008.

**Response**: The reviewer is probably correct, however we find it difficult to justify in this section since we show the data only for 2008.

P27 L27

Can the authors report the uncertainty of the retrieved emissions?

**Response**: Unfortunately, we can't calculate the uncertainties rigorously. Therefore, in estimation of uncertainties we can rely on the results of our numerical tests. We suggest that the errors in emission have similar magnitude as in the conducted tests.

P30 L5

Please indicate which BC case is used in this section.

**Response**: Thanks for your suggestion. In the posterior GEOS-Chem model simulation, we use Case 1 BC emission. We have added this information in the text. The difference of 2 cases of total BC emission is due to the different assumptions of BC particle absorbing ability. Thus, the GEOS-Chem posterior AAODs based on 2 cases of BC emissions, coupling with the 2 cases assumption of BC particle absorbing properties, show small differences (R=0.9, MAE=0.006, NMB=0.43%).

[Figure]

Figure S2. Comparison of posterior GEOS-Chem AAOD at 565nm based on 2 cases of BC emission. The correlation coefficient (R), mean absolute error (MAE), linear regression fit (Y=aX+b) and normalized mean bias (NMB) are provide in the top left corner.

Figure 18

The colour code of each square in the panels represent the number of pairs (observation, model) that fit in the square. The size of the squares are different in all the panels (in fact, some of them are not squares, despite the same limits of the x and y axes), so they are not comparable. The worst case is in the last column. The authors should write in the caption the size of the AAOD bins (and AOD for Figure 17). I strongly recommend to improve this figure.

**Response**: Good suggestion. The sizes of the AAOD bins in Figure 18 and AOD bins in Figure 17 have been added in the figure captions. And the Figure 18 has been changed to use the same size of the squares in the all panels.

P35 L22

This sentence should be written before (around line 10). Is the model sampled according to MODIS

availability?

**Response**: This sentence intended to indicate one possible reason that the posterior model AOD is a little lower than MODIS on monthly mean scale. Here, we used GEOS-5/GOCART model monthly mean value, which is averaged of all data during a month.

P36 L10-11

I could understand that these kind of measurements are more sensitive to the absorption properties, thus the retrieval of them is better constrained, but this first sentence of the paragraph implies that these are the only measurements sensitive to the absorption, which is a strong statement. Could you provide a reference on this?

**Response**: We agree, and the sentence has been revised.

P36 and P37

From panels a, b and c of Figure 21, it seems that the model/satellite comparison is not collocated. This could introduce errors in the analysis of the results, and it should be mentioned. Also, I would suggest to plot with transparent colour the missing data, and not with blue (which is equivalent to zero AAOD)

**Response**: Yes, the spatial distribution for model and satellite is not fully collocated. We use the OMI Level 3 monthly mean AAOD data (OMAERUV version 1.7.4) with a 0.5° x 0.5° spatial resolution, while the model AAOD is 2.0° x 2.5°. In panels a, b and c, we want to show the seasonal spatial pattern of AAOD from model and satellite, so we keep their best spatial resolution. In panel d and e, the grid-to-grid comparison between prior/posterior model and satellite AAOD is presented. In this comparison, the satellite AAOD is re-scaled to the model 2.0° x 2.5° resolution.

[Figure]

Figure S3. The collocated 2.0° x 2.5° seasonal AAOD from OMI (a), prior (b) and posterior (c) GEOS-5/GOCART simulation

P38 L17

Please rephrase this sentence to improve readability (It is not easy to figure out what a "800% lower" means).

**Response**: Revised.

**References**

Bennett A.F., Inverse Modeling of the Ocean and Atmosphere, Cambirge University Press, 220 p. 2002.

Bond, T. C., Doherty, S. J., Fahey, D. W., Forster, P. M., Berntsen, T., DeAngelo, B. J., Flanner, M. G., Ghan, S., Kärcher, B., Koch, D., Kinne, S., Kondo, Y., Quinn, P. K., Sarofim, M. C., Schultz, M. G., Schulz, M., Venkataraman, C., Zhang, H., Zhang, S., Bellouin, N., Guttikunda, S. K., Hopke, P. K., Jacobson, M. Z., Kaiser, J. W., Klimont, Z., Lohmann, U., Schwarz, J. P., Shindell, D., Storelvmo, T., Warren, S. G. and Zender, C. S.: Bounding the role of black carbon in the

climate system: A scientific assessment, J. Geophys. Res. Atmos., 118(11), 5380–5552, doi:10.1002/jgrd.50171, 2013.

Dubovik, O., Lapyonok, T., Kaufman, Y. J., Chin, M., Ginoux, P., Kahn, R. A. and Sinyuk, A.: Retrieving global aerosol sources from satellites using inverse modeling, Atmos. Chem. Phys., 8(2), 209–250, doi:10.5194/acp-8-209-2008, 2008.

Ginoux, P., Chin, M., Tegen, I., Prospero, J. M., Holben, B., Dubovik, O. and Lin, S.-J.: Sources and distributions of dust aerosols simulated with the GOCART model, J. Geophys. Res. Atmos., 106(D17), 20255–20273, doi:10.1029/2000JD000053, 2001.

Fairlie, T. D., Jacob, D. J. and Park, R. J.: The impact of transpacific transport of mineral dust in the United States, Atmos. Environ., 41(6), 1251–1266, doi:10.1016/j.atmosenv.2006.09.048, 2007.

Huneeus, N., Boucher, O. and Chevallier, F.: Simplified aerosol modeling for variational data assimilation, Geosci. Model Dev., 2(2), 213–229, doi:10.5194/gmd-2-213-2009, 2009.

Huneeus, N., Chevallier, F. and Boucher, O.: Estimating aerosol emissions by assimilating observed aerosol optical depth in a global aerosol model, Atmos. Chem. Phys., 12(10), 4585–4606, doi:10.5194/acp-12-4585-2012, 2012.

Huneeus, N., Boucher, O. and Chevallier, F.: Atmospheric inversion of SO2 and primary aerosol emissions for the year 2010, Atmos. Chem. Phys., 13(13), 6555–6573, doi:10.5194/acp-13-6555-2013, 2013.

Talagrand, O. and Courtier, P.: Variational assimilation of meteorological observations with the adjoint of the vorticity equations: Part I., Theory, Q. J. Roy. Meteor. Soc., 113, 1311–1328, 1987

Tarantola, A., Inverse problem theory and methods for model parameter estimation, Society for Industrial and Applied Mathematics, 2005.

Wang, J., Xu, X., Henze, D. K., Zeng, J., Ji, Q., Tsay, S.-C. and Huang, J.: Top-down estimate of dust emissions through integration of MODIS and MISR aerosol retrievals with the GEOS-Chem adjoint model, Geophys. Res. Lett., 39(8), L08802, doi:10.1029/2012GL051136, 2012.

Zender, C. S., Bian, H. and Newman, D.: Mineral Dust Entrainment and Deposition (DEAD) model: Description and 1990s dust climatology, J. Geophys. Res., 108(D14), 4416, doi:10.1029/2002JD002775, 2003.